# Cancer-associated fibroblast classification in single-cell and spatial proteomics data

Lena Cords[1,2,3,6], Sandra Tietscher[1,2,3,6], Tobias Anzeneder[4], Claus Langwieder[5], Martin Rees[5], Natalie de Souza[1,2] & Bernd Bodenmiller ◎[1,2] ✉

Cancer-associated fibroblasts (CAFs) are a diverse cell population within the tumour microenvironment, where they have critical effects on tumour evolution and patient prognosis. To define CAF phenotypes, we analyse a single-cell RNA sequencing (scRNA-seq) dataset of over 16,000 stromal cells from tumours of 14 breast cancer patients, based on which we define and functionally annotate nine CAF phenotypes and one class of pericytes. We validate this classification system in four additional cancer types and use highly multiplexed imaging mass cytometry on matched breast cancer samples to confirm our defined CAF phenotypes at the protein level and to analyse their spatial distribution within tumours. This general CAF classification scheme will allow comparison of CAF phenotypes across studies, facilitate analysis of their functional roles, and potentially guide development of new treatment strategies in the future.

The tumour microenvironment (TME) is a complex ecosystem consisting of diverse and interacting cell populations. These cell populations include a variety of resident and infiltrating immune cells and stromal cells, as well as tumour cells. The composition of the TME influences tumour progression and metastasis[1], the anti-tumour immune response[2], and therapy response[3]. It is thus crucial to study tumours as complex ecosystems in order to understand intercellular interactions and to improve patient prognosis.

Fibroblasts are the main constituent of the tumour stroma. Cancer-associated fibroblasts (CAFs) are diverse cells with numerous roles within the TME[4]. They are key players in shaping the tumour microenvironment with functions in tumour promotion[5,6] and inflammation[7–9] as well as maintenance and reshaping of the extracellular matrix (ECM)[4,10]. Different CAF subpopulations have been described in various cancer types[11–14]. In breast cancer, for example, single-cell RNA sequencing (scRNA-seq) in a mouse model detected four different CAF phenotypes, termed vascular CAFs, matrix CAFs, cycling CAFs and developmental CAFs[15]. A separate scRNA-seq study using a triple-negative breast cancer mouse model further identified CAF phenotypes with inflammatory and immune regulatory, ECM-

producing, protein folding, and antigen-presenting functions[16]. In human breast cancer, the CAF-S1 subtype is associated with an immunosuppressive environment[13] and cancer cell migration, thus likely promoting metastasis[17].

CAF subsets have been identified and linked to patient prognosis or therapy response[18] in many cancer types including, for example, pancreatic ductal adenocarcinoma (PDAC)[7,9], bladder urothelial cancer[14], melanoma[19] and lung cancer[11,18], highlighting their importance in cancer research. While CAF heterogeneity was initially typically studied in a single cancer type[18,20,21], cross-species[7,9,12,22] and cross-cancer[22–24] validation has become easier with increased data availability from published studies. However, different studies may vary in their biological focus and identify CAF types at different phenotypic granularity, such that they are not always easily comparable. Thus, despite recent advances, a simple and decisive set of markers for unambiguous identification of CAFs and CAF subtypes in the TME would be very useful[10]. To define such a marker set, we set out to establish a tumour-type-independent CAF classification system.

In this work, we analysed a scRNA-seq dataset of human breast cancer and defined nine CAF and one pericyte populations (Fig. 1). We

[1]Department of Quantitative Biomedicine, University of Zurich, CH-8057 Zurich, Switzerland. [2]Institute of Molecular Health Sciences, ETH Zurich, CH-8093 Zurich, Switzerland. [3]Life Science Zurich Graduate School, ETH Zurich and University of Zurich, CH-8057 Zurich, Switzerland. [4]Patients' Tumor Bank of Hope (PATH), D-81337 Munich, Germany. [5]Pathology at Josefshaus, D-44137 Dortmund, Germany. [6]These authors contributed equally: Lena Cords, Sandra Tietscher. ✉e-mail: bernd.bodenmiller@uzh.ch

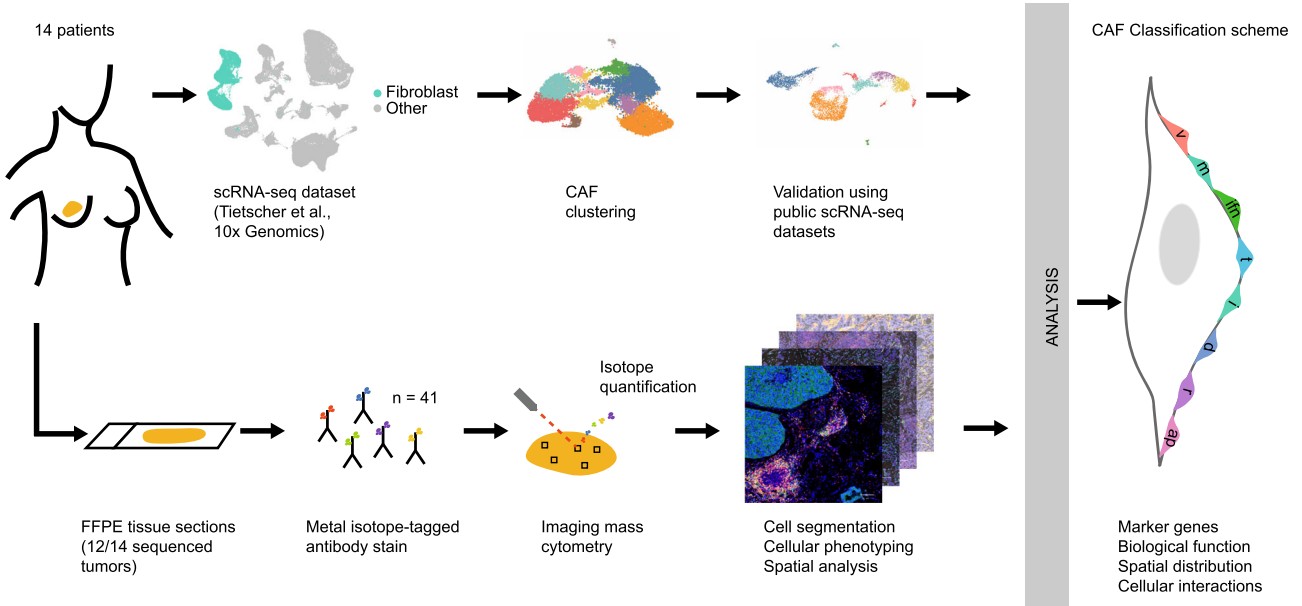

**Fig. 1 | Workflow used to define the CAF classification system.** scRNA-seq data from matched breast cancer samples[25] were analysed for fibroblast heterogeneity with subsequent validation of identified fibroblast subtypes in other tumour types. IMC using formalin-fixed paraffin-embedded tissue sections (FFPE) was used to validate findings at the protein level and to evaluate spatial distribution. The resulting CAF classification system (featuring vascular CAFs (vCAFs), matrix CAFs (mCAFs), interferon-response CAFs (ifnCAFs), tumour-like CAFs (tCAFs), inflammatory CAFs (iCAFs), dividing CAFs (dCAFs), reticular-like CAFs (rCAFs) and antigen-presenting CAFs (apCAFs)) was based on marker genes, biological functions, spatial distribution within the TME, and cellular interactions.

confirmed these populations at the protein level using multiplex imaging mass cytometry (IMC) on matched samples, defined marker genes for each CAF type, and investigated their spatial distribution within the TME such as distance to the stroma/tumour border, to structures such as vessels, and to classes of neighbouring cells (Fig. 1). Our analysis across several tumour types showed that these cellular subpopulations are present across cancer types and thus that the CAF phenotypes we have defined can be generalised. Our proposed classification system for CAFs should enable future work by providing suggested marker genes for generally identifiable and functionally interpretable CAF types.

## Results

### Fibroblast classification in breast cancer

To identify and classify CAF types, we analysed our previously generated scRNA-seq dataset from 14 human breast cancer specimens (Supplementary Data 1), in which we had identified 16,704 stromal cells (out of ~119,000 cells total) based on unsupervised clustering[25]. Of these stromal cells, we identified 2,389 cells as pericytes based on the expression of *RGS5*[26], resulting in a total of 14,315 CAFs. We used this dataset to investigate fibroblast phenotypic heterogeneity in breast tumours.

After batch correction (Supplementary Fig. 1a), we performed unsupervised hierarchical clustering of the single-cell gene expression profiles of all stromal cells, which identified 12 clusters at a resolution of 0.4 (Supplementary Fig. 1b, c and Supplementary Data 2). This resolution was selected to balance between overclustering and retaining the ability to identify rare cell types. We then examined the top differentially expressed genes (MAST[27]) for each cluster relative to all other clusters (Fig. 2a, b). In two instances, we manually merged two clusters based on the high overlap of functionally related, differentially expressed genes (see "Discussion"), resulting in a total of 10 cell types, and repeated the differential gene expression analysis after cluster merging (Supplementary Data 3). Gene set enrichment analysis was then used to identify hallmark pathways[28] enriched for each cell type (Fig. 2c). Based on these

analyses, we annotated the clusters as nine CAF types and one cluster of pericytes:

Matrix CAFs (mCAFs): We identified two clusters (clusters 0 and 7), characterised by high levels of expression of genes encoding matrix proteins, in particular matrix metalloproteinase *MMP11* and *COL1A2* (Fig. 2a, Supplementary Fig. 1b and Supplementary Data 2 and 3), which we combined into a single cluster. The resulting group of CAFs (*n* = 4525 cells) expressed *MMP11* and other matrix metalloproteinase-encoding mRNAs, and collagen-encoding mRNAs (*COL10A1, COL11A1, COL8A1, COL1A2, COL12A1, COL3A1, COL8A1* and *COL5A2*). Further, mRNAs encoding non-collagenous matrix proteins (*COMP* and *POSTN*) were also amongst the top 20 differentially expressed genes of this group (Fig. 2a, b). Cells of this cluster also expressed high levels of genes associated with adhesion (e.g., *LRRC15, LRRC17,* and *ASPN*) and with migration (e.g., *POSTN, SULF1, INHBA* and *VCAN*) (Fig. 2a and Supplementary Data 2 and 3). As the top two differentially expressed genes of this cluster were *MMP11* and *POSTN*, we named these cells mCAFs, also in alignment with prior studies[15,16,24]. Despite the strong overlap in their top differentially expressed genes, it remains possible that the two merged clusters (0 and 7) represent distinct subgroups of mCAFs; in particular the top differentially expressed genes exclusive to the smaller cluster 7 (*MGP* and *BGN*) could suggest an association with cartilage. Next, we conducted a gene set enrichment analysis (GSEA) using hallmark pathways to assess our annotations in an orthogonal manner[28,29]. For mCAFs, the pathways upregulated included TGF-β signalling, which is associated with the development of activated myofibroblasts[30], KRAS signalling, and pathways underlying the epithelial to mesenchymal transition (EMT) (Fig. 2c). The EMT pathway is defined, in this analysis, by multiple collagens, as well as *CTHRC1, FAP, INHBA, LRRC15, MGP, POSTN* and *VCAN*. These are all genes associated with matrix remodelling and migration, supporting our mCAF annotation.

Inflammatory CAFs (iCAFs): Cells belonging to the second largest cluster (*n* = 3439 cells) were characterised by unique expression of a phospholipase encoded by *PLA2G2A* and of genes involved in the complement pathway (e.g., *CFD* and *C3*) (Fig. 2a, b). In addition, *CD34*

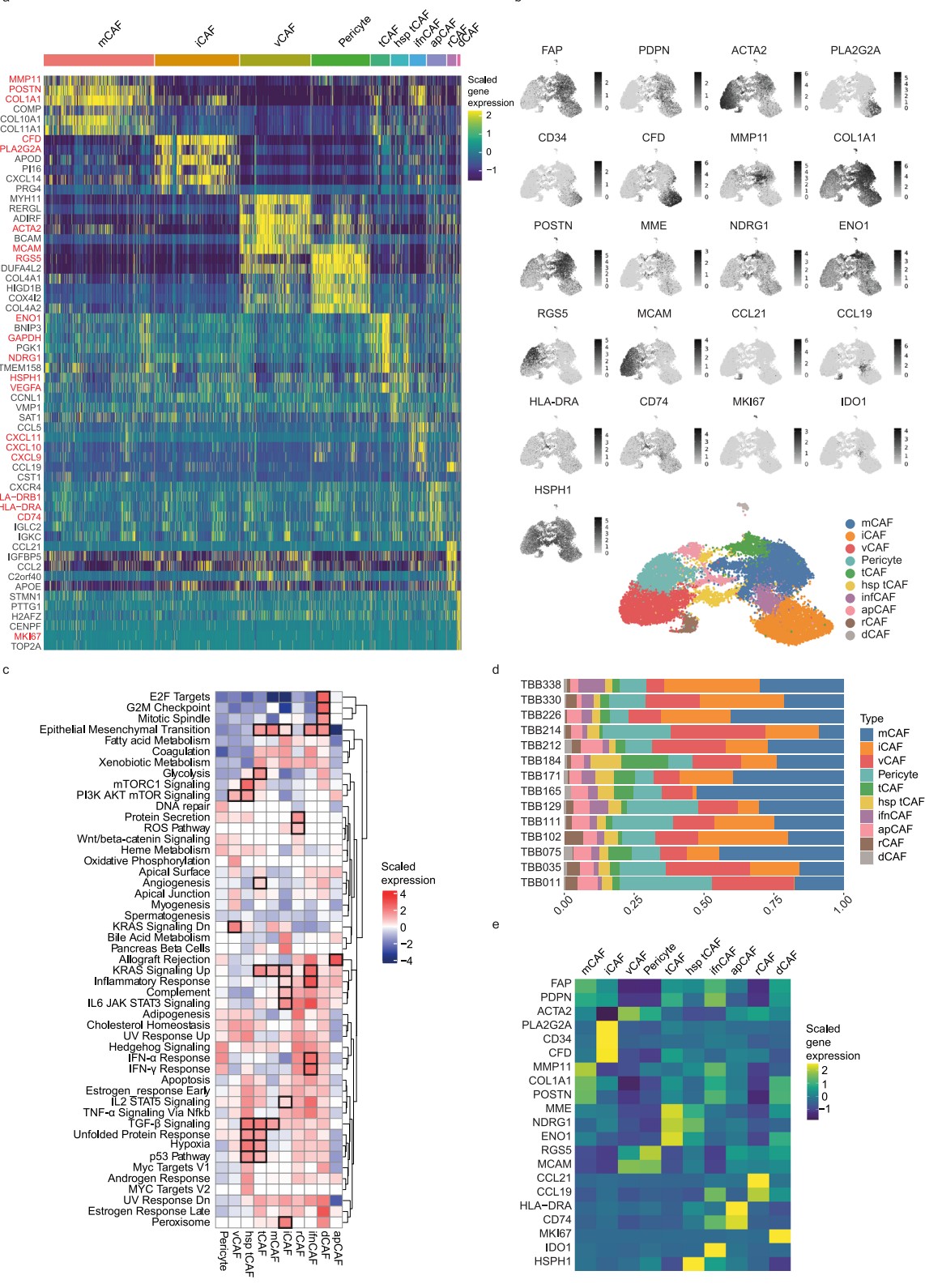

**Fig. 2 | Fibroblast heterogeneity in breast cancer. a** Heatmap of the top six differentially expressed genes for each cell type in scRNA-seq data of all stromal cells (*n* = 16,704). Cellular phenotypes are indicated above the heatmap. Key marker genes are highlighted in red. **b** UMAP of all stromal cells coloured by CAF type, together with the corresponding feature plots showing the expression level of selected marker genes for each cell. **c** Gene set enrichment analysis comparing the enrichment of hallmark pathways between CAF types. Boxes indicate the functional hallmark pathways that we used to define/annotate CAF types (see also Supplementary Data 7). **d** Proportion of all CAF types and pericytes per patient. **e** Heatmap showing the average gene expression level of all defined marker genes per identified cell type (after batch correction).

was among the top differentially expressed genes in this cluster (Supplementary Data 2 and 3). *CD34* is a marker for hematopoietic stem cells but is known to be expressed by fibroblasts and has previously been linked to inflammation[12,21]. This cluster also showed high expression of cytokines and chemokines, including *CXCL12*, *CXCL14* and *IL6* (Supplementary Data 2 and 4). *CXCL12* has previously been used as a marker for CAFs associated with inflammation[14,21] and for immune regulatory CAFs[16], and we thus labelled this cluster iCAFs. Consistent with pro-inflammatory activities, the GSEA revealed that iCAFs are characterised by an upregulation of the IL6-JAK-STAT3 pathway as well as KRAS and complement signalling (Fig. 2c). The pancreatic beta cell pathway also showed strong enrichment in iCAFs which is most likely due to a high gene expression level of *DPP4* in this cluster (Supplementary Data 2 and 3).

Vascular CAFs (vCAFs): The overall gene expression profile of the CAF cluster with the third highest number of cells (*n* = 2886 cells) suggested a link to angiogenesis since these cells showed high expression of *NOTCH3*, an important receptor in vascularisation and angiogenesis[31], of collagen *COL18A1*, involved in angiogenesis regulation[32], and of *MCAM* (which encodes CD146) (Fig. 2a, b and Supplementary Data 2 and 3). A fraction of these cells showed expression of pericyte marker *RGS5* (Fig. 2a, b and Supplementary Data 2 and 3), but *RGS5* was not among the top differentially expressed genes of this cluster (Supplementary Data 2 and 3). We named these cells vCAFs, in line with other descriptions of CAFs showing pro-angiogenic features characterised in human and mouse cancer[15,21]. While vCAFs did show strong gene expression overlap with *RGS5*⁺ pericytes (e.g., *MCAM* and *ACTA2*) (*n* = 2389 *RGS5*⁺ pericytes), there were clear differences between these cell types as well; for instance, pericytes did not show strong expression of *RERGL* and *MHY11*.

Tumour-like CAFs (tCAFs): We observed strong differential expression of proliferation-, migration- and metastasis-associated genes (e.g., *PDPN*, *MME*, *TMEM158* and *NDRG1*) as well as stress-response-associated genes (e.g., *ENO1*), and *GAPDH* in a small cluster (*n* = 786 cells) (Fig. 2a, b and Supplementary Data 2 and 3). This cluster uniquely expressed high levels of *MME* (which encodes CD10), a membrane metalloprotease, and *TMEM158*, an indicator of Ras pathway activation, and also expressed high levels of *VEGFA*, which promotes angiogenesis and vascularisation. Since this gene expression signature resembles that of tumour cells, we named this cluster tCAFs; cells with this phenotype have been previously associated with chemoresistance[18]. This cluster also showed elevated expression of hypoxia marker carbonic anhydrase IX (*CAIX*, Supplementary Data 2 and 3), suggesting proximity to tumour-derived hypoxic regions.

A second cluster of tCAFs (*n* = 722) expressed high levels of transcripts encoding heat-shock proteins including *HSPH1* and *HSP90AA1* (Fig. 2a, b and Supplementary Data 2 and 3), as well as most of the tCAF-characteristic genes. We interpret these cells as tCAFs that are under higher levels of cellular stress and labelled them heat-shock protein-high tCAFs (hsp_tCAFs). Although the stress phenotype was not identified in the GSEA, it did show an upregulation in both tCAF clusters of numerous pathways including EMT-associated pathways, TGF-β and KRAS signalling, as well as glycolysis, MTORC1, and PI3K/Akt/mTOR signalling, the P53 pathway, and hypoxia (Fig. 2c). Since many of these upregulated pathways are also seen in tumour cells, this is consistent with our annotation of these cells as tumour-like CAFs.

Interferon-response CAFs (ifnCAFs): We annotated a cluster (*n* = 655) with high differential expression levels of genes associated with chronic inflammation (e.g., *IL32*), as well as of genes upregulated in response to interferons (e.g., *CXCL9*, *CXCL10*, *CXCL11* and *IDO1*) as ifnCAFs, due to their strong interferon response (Fig. 2a, b and Supplementary Data 2 and 3). This is consistent with previous work that has identified CAFs secreting IDO, affecting the immune compartment[33]. Further, the GSEA of this cluster showed a strong upregulation of the inflammatory response pathways and of

interferon-α and interferon-γ responses (Fig. 2c), although several additional signalling pathways were enriched in these cells (e.g., IL2-STAT5, TNF-α, IL6-JAK-STAT3 and KRAS signalling).

Antigen-presenting CAFs (apCAFs): We identified two clusters (clusters 8 (*n* = 427) and 10 (*n* = 366), Supplementary Fig. 1b and Supplementary Data 2 and 3) with high expression of genes involved in MHC-II-associated antigen presentation, including *HLA-DRA*, *HLA-DRB1* and *CD74* (Fig. 2a, b). We manually grouped these clusters based on strongly overlapping top differentially expressed marker genes that also shared a biological function (i.e., antigen presentation), and named the resulting cluster apCAFs, as previously suggested[12]. In the GSEA, apCAFs showed the highest upregulation amongst all CAF types of the allograft rejection pathway, in line with their strong expression of MHC-II machinery-related genes (Fig. 2c). Despite the strong relatedness of the merged clusters 8 and 10, we cannot rule out that they represent different subgroups of apCAFs.

Reticular-like CAFs (rCAFs): A relatively rare cluster (*n* = 373 cells) showed strong differential expression of *CCL21* and *CCL19* (Fig. 2a, b and Supplementary Data 2 and 3). These are markers of reticular fibroblasts in lymphoid tissues that facilitate homing of naïve T cells[34], and we named the cells accordingly.

Dividing CAFs (dCAFs): The smallest cluster of CAFs (*n* = 126 cells) in our dataset displayed high expression of genes upregulated during cell division (e.g., *TUBA1B* and *MKI67*, Fig. 2a, b and Supplementary Data 2 and 3); these have been previously defined as cycling CAFs[15]. The GSEA supported annotation as dCAFs, showing that E2F targets, the G2M checkpoint, and mitotic spindle pathways were upregulated in cells of this cluster (Fig. 2c).

To summarise, differential gene expression analysis together with gene set enrichment analysis identified 10 biologically interpretable cell types (9 CAF types and one cluster of pericytes) with unique gene expression profiles. All phenotypes, even rare ones such as rCAFs, were detected in each of our 14 patients (Fig. 2d). Further, the cells could broadly be separated into two groups (Fig. 2b, e, Supplementary Fig. 1d and Supplementary Data 4). Among other markers, these groups could be distinguished by their expression of *FAP* and *SMA*, which have been used in previous studies to define myofibroblasts or CAFs in general and to preselect CAFs before sequencing[7,13,30,35]. *FAP*⁺ CAFs have previously been defined as myofibroblast-like[30]; this group includes the mCAF, iCAF, tCAF, hsp_tCAF, ifnCAF, apCAF, dCAF and dCAF clusters. The *FAP*⁻ CAFs include vCAFs, pericytes, and rCAFs (Fig. 2e). Batch correction had no substantial effect on the recovered CAF types (Supplementary Fig. 1e).

To enable a straightforward identification of our defined CAF types using different methods, for example, flow cytometry or imaging, we sought to define the subset of marker genes that best identified each CAF type. To this end, we examined the top differentially expressed genes for each CAF type and selected those markers that were both in agreement with the current literature[10,13,15,16] and also compatible with our goal of conducting a spatial imaging analysis of CAF types within tumour tissue (i.e., proteins with high-quality antibodies for multiplex imaging that should not be confounded with markers for other non-stromal cell types). Our selected markers were *RGS5* for pericytes, *MCAM* for vCAFs, *MME* (together with *NDRG1* and *ENO1*) for tCAFs, *IDO1* for ifnCAFs, *HSPH1* for hsp_tCAFs, *MMP11* (together with *COL1A1* and *POSTN*) for mCAFs, *PLA2G2A*, *CDF* and *CD34* for iCAFs, *MKI67* for dCAFs, *CCL21* (and *CCL19*) for rCAFs, *CD74* and *HLA-DR* for apCAFs and *PDPN* and *FAP* for the general category of activated CAFs (Fig. 2e). When cells from the scRNA-seq dataset were clustered using only these marker genes, we were able to distinguish all our proposed CAF phenotypes (Supplementary Fig. 1f, g). These markers were then used to select markers for subsequent imaging. In summary, we identified nine CAF types and a single group of pericytes in tumours from a cohort of 14 breast cancer patients and identified a set of 17 markers to effectively distinguish between these CAF types.

## A general CAF classification scheme

To test whether our classification scheme is independent of tumour type, we next investigated CAF heterogeneity in four publicly available scRNA-seq datasets from non-small cell lung cancer (NSCLC)[11], colon cancer[36], pancreatic ductal carcinoma (PDAC)[37] and head-and-neck squamous cell carcinoma cancer (HNSCC)[38]. In the lung cancer dataset, we increased the number of fibroblasts by pooling raw data of both the test and validation cohorts from the previous study, resulting in 1377 CAFs as defined by our criteria and 574 cells from adjacent healthy tissue (Supplementary Fig. 2). The colon cancer dataset contained 1568 CAFs originating from tumour and 1917 cells from healthy tissue (Supplementary Fig. 3). The PDAC dataset contained 1762 fibroblasts from tumour tissue (Supplementary Fig. 4). The HNSCC dataset contained 1354 cells that we identified as fibroblasts (Supplementary Fig. 5); of those, 1016 fibroblasts originated from tumour tissue, and 315 came from matching lymph node metastases. Taken together, our integrated dataset contained 5723 CAFs from four different primary tumour types (Fig. 3a); we excluded cells from healthy tissue and metastatic sites to keep the integrated dataset comparable to our breast cancer dataset, which only included cells from primary tumours. The datasets were integrated using anchors to correct for batch effects as previously described[39], followed by further batch correction as was performed for the breast cancer dataset (Fig. 3a).

We used a two-step approach to identify CAF types in the integrated dataset and to validate our findings. We first performed unbiased clustering on full single-cell gene expression profiles of the integrated dataset and identified all our previously defined CAF types as well as pericytes (Fig. 3b, c, Supplementary Fig. 6a, b and Supplementary Data 5 and 6). All CAF types were detected in all cancer types (Fig. 3d). Analysis of this integrated dataset yielded similar top differentially expressed genes for each CAF type as identified in the breast cancer dataset (Fig. 3b, c, e and Supplementary Data 5 and 6), and GSEA analysis showed that the top five enriched pathways in the breast cancer dataset were also enriched in the integrated dataset in 60–80% of cases (Supplementary Data 7). Although imperfect, we saw a particularly strong overlap in the pathways that informed our functional annotation of CAF types (Fig. 3f and Supplementary Data 7). Specifically, the top five enriched pathways in the integrated dataset included hypoxia in tCAFs, the EMT pathway and KRAS signalling in mCAFs, IL6-JAK-STAT signalling in iCAFs, allograft rejection in apCAFs, and E2F targets and the G2M checkpoint in dCAFs, all of which overlapped with top enriched pathways in the breast cancer dataset (Fig. 3f). Clustering the integrated dataset using only our breast cancer-defined CAF marker genes (Fig. 3e) also identified all our defined CAF types (Supplementary Fig. 6c–e), providing further support that these markers are sufficient to classify CAFs.

We also clustered each validation dataset individually (Supplementary Figs. 2–5). We detected most CAF types in each tumour dataset in these analyses, although dCAFs were typically missing, and vCAFs and pericytes could not be well distinguished in the colon cancer and PDAC datasets. Further, the PDAC dataset was dominated by mCAFs and iCAFs when analysed individually, consistent with the known fibrotic nature of this type of cancer[12,20,37]. Analysis of a larger PDAC dataset (GSE212996) with over 7000 CAFs showed that, although heavily dominated by mCAFs and iCAFs, tCAFs, apCAFs and vCAFs were detected (Supplementary Fig. 7). Given the high number of cells in the large PDAC dataset compared to the other validation datasets, we did not include it in the integrated analysis as it could potentially drive the clustering results. In our analyses of the individual datasets, we also included cells from healthy adjacent tissue or metastatic lesions to assess the specificity of the CAF types to primary tumour tissue. All CAF types were detected at all tissue sites but in different numbers (Supplementary Figs. 2d, 3 and 4d). For example, tCAFs were almost exclusively found in primary tumours (lung: 262

cells in tumour, 11 cells in healthy tissue; colon: 174 cells in tumour, 1 cell in healthy tissue; HNSCC: 284 cells in tumour, 13 cells in metastatic lymph node; two-sided $t$ test, $P = 0.019$). In summary, we were able to identify breast cancer-defined CAF types across multiple cancer types, suggesting that we have defined a general CAF classification system and marker genes.

## Spatial distribution of CAF phenotypes in breast tumours

We next used multiplexed IMC[40] to analyse the spatial distributions of our defined CAF phenotypes in breast cancer samples from the same patients as were analysed with scRNA-seq. The IMC analysis also allowed us to validate our findings at the protein level. We used our scRNA-seq data to guide design of a 41-plex antibody panel to discriminate the different CAF phenotypes (Fig. 4a and Supplementary Table 1). We used expression of FAP and PDPN to distinguish myofibroblasts from FAP- CAFs. We defined vCAFs as CD146high, CD34negative; and iCAFs as CD34high, CD146+), note that CD146 (encoded by *MCAM*) expression in iCAFs is lower than in vCAFs. We used CD10 (encoded by *MME*) for tCAFs, CCL21 for rCAFs, IDO to identify ifnCAFs, and the proliferation marker Ki-67 for dCAFs. Due to the lack of an antibody suitable for the detection of MMP11, mCAFs were identified as PDPNlow/collagen-fibronectinhigh/FAP+ cells that did not express CD10, CD34 or CD146. Finally, we did not identify apCAFs in IMC because HLA-DR is highly expressed by myeloid cells in the stroma, causing high signal overlap. The IMC antibody panel also included markers that allowed the identification of tumour cells and immune cells with a focus on T-cell subtypes (Fig. 4a and Supplementary Fig. 8a). Clustering of the scRNA-seq data using the CAF-targeted subset of the 41 markers in this IMC panel showed that we could recover all our defined CAF types (Supplementary Fig. 8b).

We stained 12 breast tumour samples (matched tissue samples for 12 of the 14 patient samples analysed by scRNA-seq) with our 41-plex IMC antibody panel and detected stromal, tumour, and immune cells (Supplementary Fig. 8c). We selected stroma-rich areas and areas with tertiary lymphoid structures (TLS) based on immunofluorescence imaging (Supplementary Fig. 8d), and then analysed these selected areas (7–13 per patient, depending on the number of visible TLS) with IMC. After single-cell segmentation (Supplementary Fig. 8e), we identified a total of 222,318 tumour, 104,767 immune, and 140,999 CAFs as well as 29,635 endothelial cells and 55,402 other cells. We identified most of the scRNA-seq-defined CAF subtypes in the IMC dataset (Fig. 4b, c). We detected mCAFs, iCAFs, vCAFs, hypoxic and non-hypoxic tCAFs, ifnCAFs, and rCAFs, but not dCAFs. Most patient samples included multiple CAF types (Fig. 4d). Due to the lack of an antibody staining for RGS5, we did not distinguish between vCAFs and pericytes and labelled all CD146+/CD31-/vWF- cells as vCAFs. We also identified a cluster of CAFs that did not show expression of FAP while still showing high expression of SMA and PDGFR-b and collagens; we labelled these cells SMA CAFs.

To study the spatial distributions of CAF types, we conducted a neighbourhood analysis in which we quantified the cell types within a radius of 30 μm of a given cell type, across all imaged regions. vCAFs showed the highest mean interaction score with endothelial cells (Fig. 4e) and were exclusively found around endothelial cells in vessel-like structures in the images (Fig. 4c). We observed that rCAFs were often found in images with TLS-like structures, where they surrounded aggregated immune cells (mainly CD20+ B cells) (Fig. 4c) and they further showed a trend towards enrichment in images with TLS structures (Fig. 4f). The neighbourhood analysis showed that iCAFs neighbour both vCAFs and endothelial cells (Fig. 4c, e). Among CAF types, only CD10+/CD73+ tCAFs and ifnCAFs showed positive interaction scores with tumour cells in the neighbourhood analysis, suggesting proximity to the tumour (Fig. 4e). We investigated this further by comparing the distances of all CAF types to the tumour-stroma border. ifnCAFs were closest to tumour cells with a median

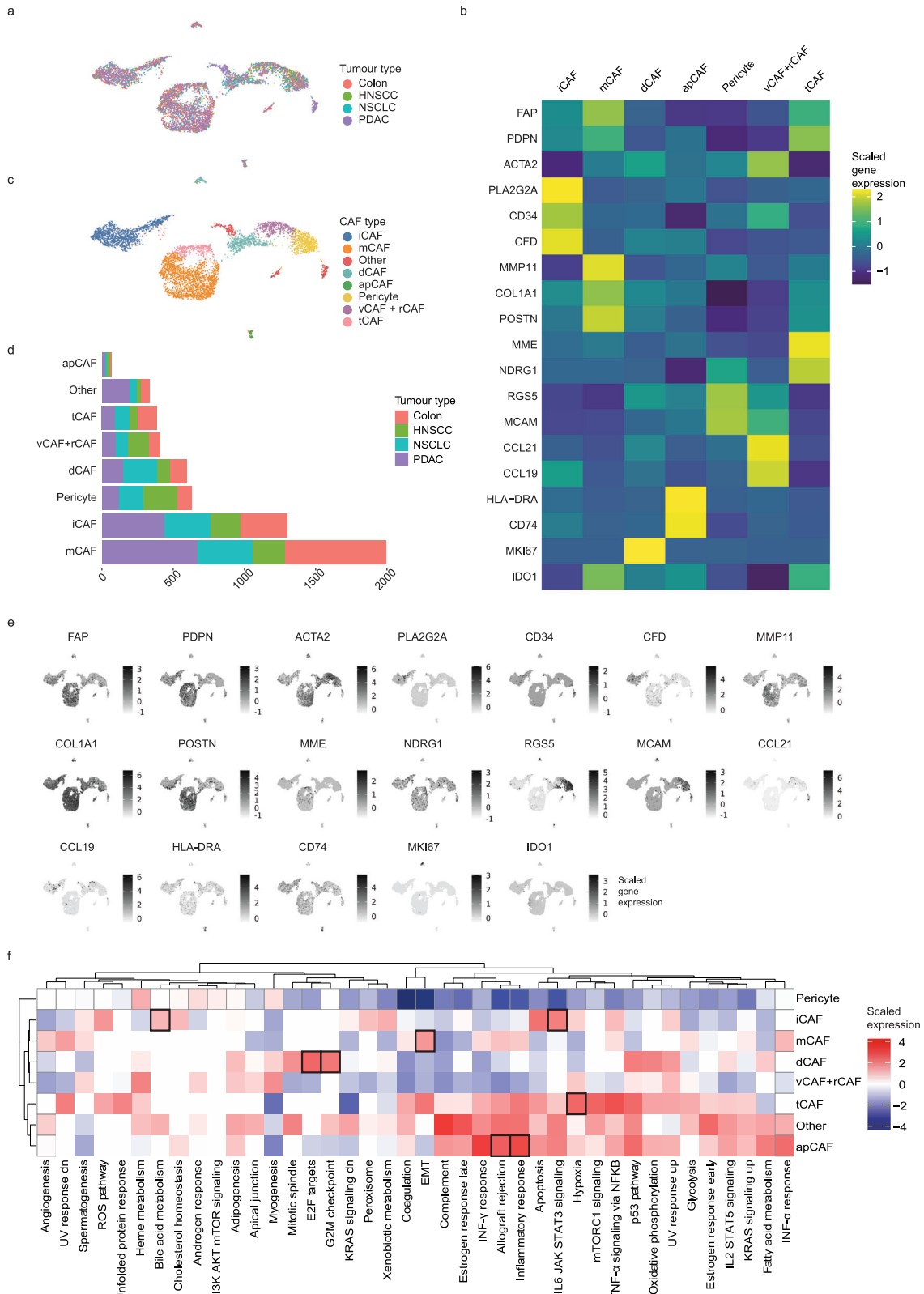

**Fig. 3 | Fibroblast heterogeneity in multiple cancer types. a** UMAP showing the validation datasets (non-small cell lung cancer (NSCLC), head-and-neck squamous cell carcinoma (HNSCC), colorectal cancer, pancreatic ductal carcinoma (PDAC)). **b** Heatmap showing the average marker gene expression of each identified cell type in the integrated validation dataset. **c** UMAP showing the final CAF classification of the validation cohort. **d** Bar chart showing the absolute numbers of all CAF types and pericytes as detected with unbiased clustering of the validation dataset and the respective proportions of each cell type per tumour type. **e** Feature plot showing the cellular expression levels of selected marker genes on the UMAP. **f** Heatmap showing the results of gene set enrichment analysis for all defined cell types. Boxes indicate the overlap of the top five enriched hallmark pathways between the integrated validation and breast cancer dataset.

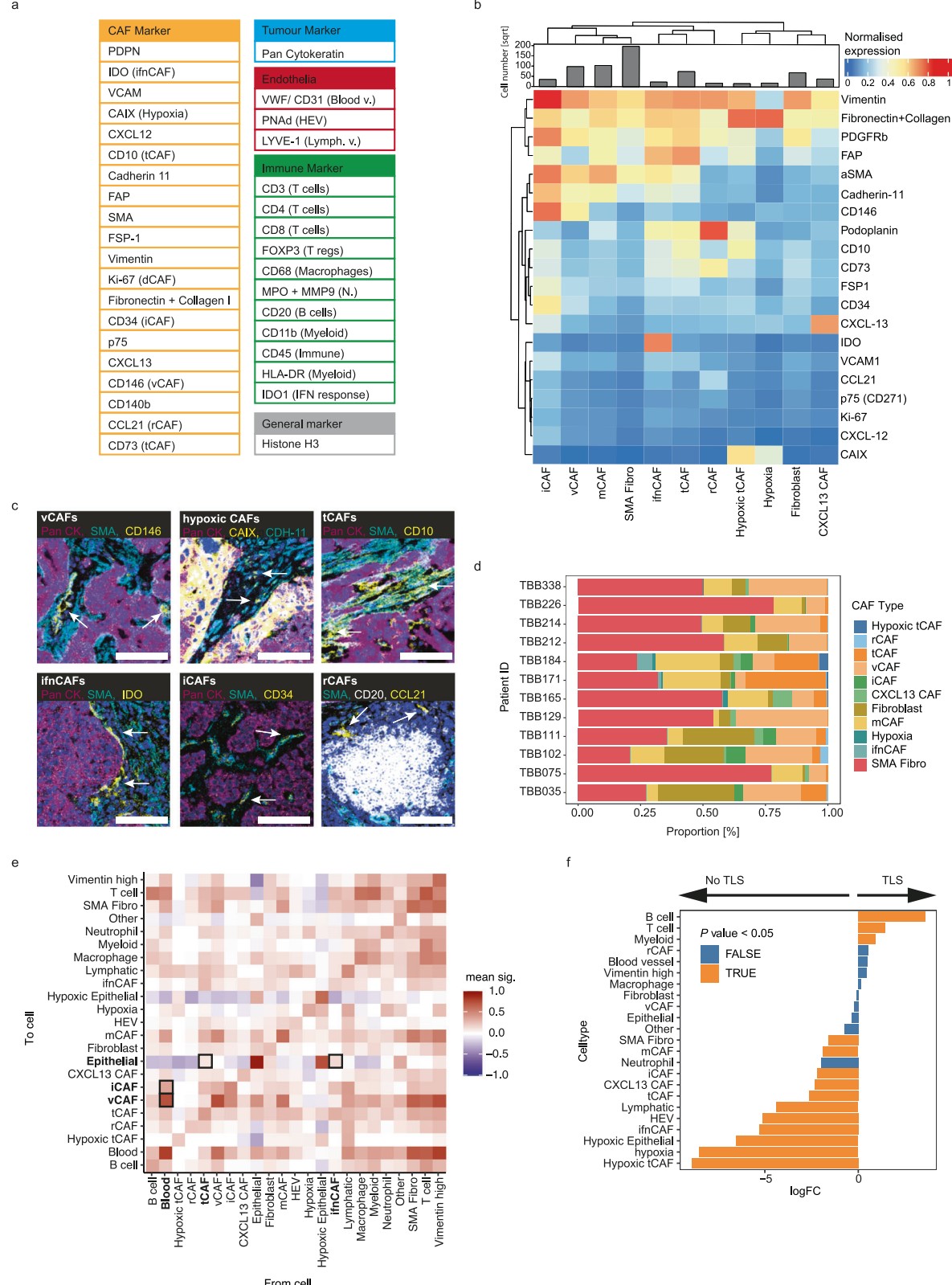

distance of 4 μm followed by tCAFs with a median distance of 14 μm to the tumour-stroma border (Supplementary Fig. 8f), thus confirming that these two CAF types are in relatively close proximity to tumour cells.

In summary, in addition to showing spatial distributions of various CAF types (Fig. 5) within breast tumours, this IMC analysis has further validated our scRNA-seq-based CAF classification system and showed that most CAF types can be identified in imaging data with only 2–5 markers per type (Table 1). If pericytes are to be distinguished, three additional markers are required. Overall, our demonstration that this classification scheme identifies biologically interpretable CAF phenotypes in multiple cancer types suggests that it will be useful as a framework for future investigations of fibroblast biology.

**Fig. 4 | Spatial analysis of CAF types in breast tumours using imaging mass cytometry. a** Panel of all markers used in the IMC study (Blood/Lymph v. = blood/lymph vessel, N. = neutrophil, HEV = high-endothelial venules). **b** Heatmap of marker expression of CAF clusters defined by IMC in breast tumour samples. The histogram indicates the square root of all cell numbers per cluster in each CAF type. **c** Zoomed-in images acquired with IMC showing the expression of key markers used in our classification system on the image level. The indicated CAF type is highlighted by arrows. CAFs are identified as follows: vCAFs, CD146; hypoxic CAFs: CDH-11, CAIX; tCAFs, SMA, CD10; ifnCAFs: IDO, SMA; iCAFs: aSMA, CD34; rCAFs: CCL21. PanCK indicates tumour cells, CD20 indicates B cells, Iridium (blue)

indicates nuclei in all images. Scalebar, 100 μm. **d** Proportion of all CAF types defined by IMC over all patients. **e** Neighbourhood analysis showing cell-to-cell interactions at the image level, over all images in the study. The cell-to-cell interactions are compared against a random null distribution using permutation testing. An interaction score is then generated for each cell pair based on the *P* values calculated on the image level (two-sided permutation test). Positive interaction scores mean that a given pair of cells is neighbouring significantly more often than compared to the null distribution. **f** Differential abundance analysis comparing cellular enrichment in TLS containing images versus images not containing any TLS-like structures.

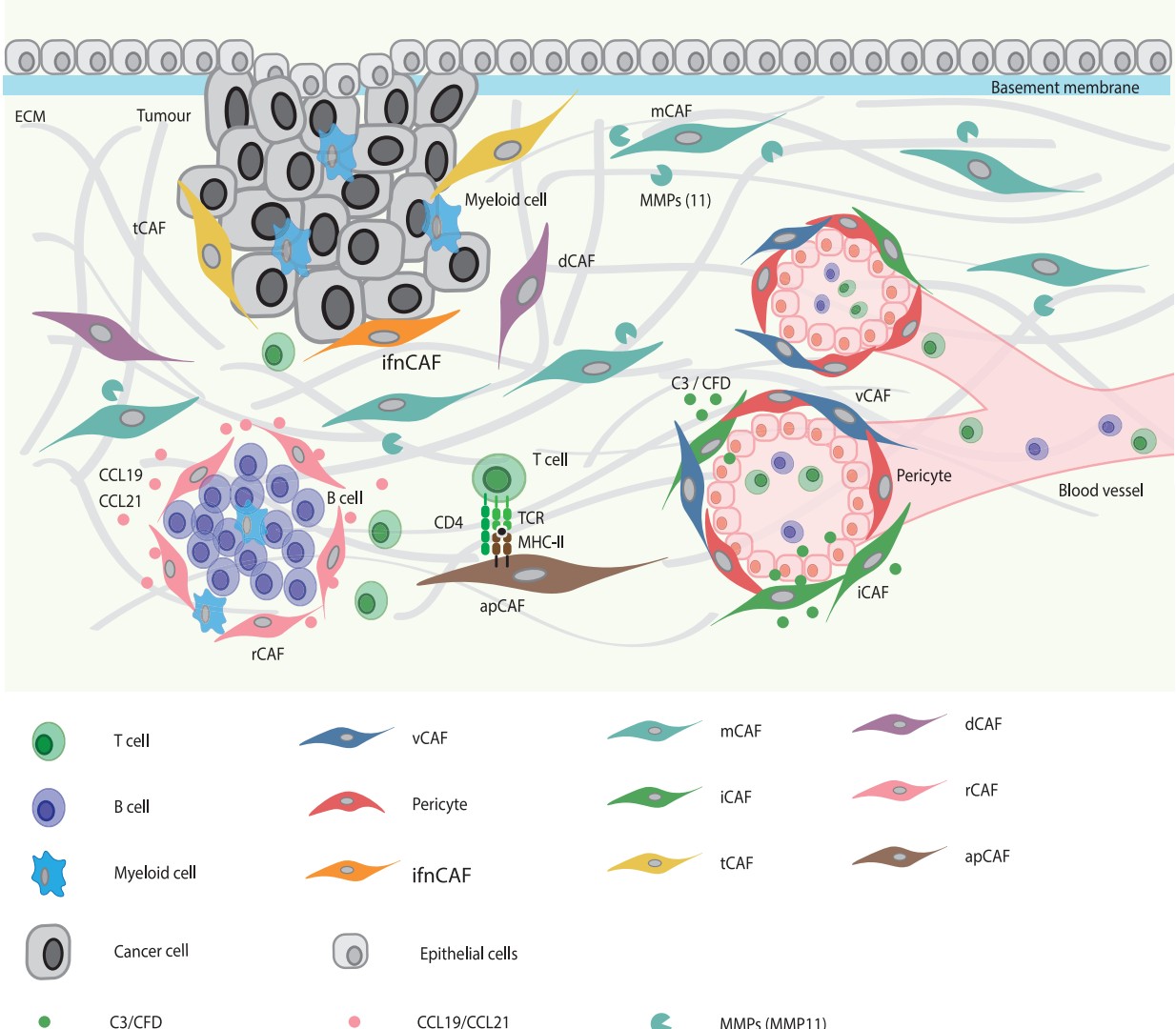

**Fig. 5 | CAF classification scheme.** Graphical summary showing the spatial distribution patterns of the seven CAF types (excluding apCAFs) in the TME as detected with imaging mass cytometry and their interaction with other cell types.

## Discussion

Studies of CAF heterogeneity in many cancer types have led to the definition of numerous CAF subtypes[12–14,16,21]. While early work typically defined CAF phenotypes in single cancer types[11,13–15,21,41], recent studies have examined pan-cancer CAF heterogeneity and have found similarities between CAFs in different cancer types[20,22–24]. Nevertheless, there remains the need for a general CAF classification system that allows for simple comparison of CAF types between studies[10]. Here, we propose such a CAF classification system based on the literature-driven annotation of CAF types in scRNA-seq data,

provide a set of markers for these CAF types, and use multiplex imaging to analyse their spatial distribution and cellular interactions in breast tumours. Based on analysis of scRNA-seq data of more than 16,000 stromal cells from 14 human breast cancer tumour samples, coupled with multiplex single-cell imaging of matched samples, we identified marker expression and spatial distribution of nine CAF types as well as pericytes. We further identified these CAF phenotypes in four additional cancer types based on previously published scRNA-seq data, suggesting that these phenotypes and the classification system are generalisable.

**Table 1 | Summary of CAF classification scheme**

| CAF type | Matrix CAFs | Inflammatory CAFs | Vascular CAFs | Tumour-like CAFs | Interferon-response CAFs | Antigen-presenting CAFs | Reticular-like CAFs | Dividing CAFs |
|---|---|---|---|---|---|---|---|---|
| **Marker genes** | *MMP11, CDH11, POSTN, Collagens* | *PLA2G2A, CFD, C3, CD34, CXCL12, CD248* | *ACTA2, MCAM, RGS5⁻* | *PDPN, MME, ENO1, NDRG1, CA9 HSPH1* (hsp_tCAFs) | *IDO, Collagens* | *CD74, HLA-DR, ACTA2* | *CCL21, CCL19, ACTA2* | *MKI67* |
| **Anticipated biological function** | Stroma and ECM maintenance | Immune cell infiltration, inflammation | Support of angiogenesis, vascularisation | Tumour growth, angiogenesis | IFN response | Antigen presentation via MHC-II | TLS formation, T cell attraction | Cell division |
| **FAP expression** | FAP⁺ | FAP⁺ | FAP⁻ | FAP⁻ | FAP⁺ | FAP⁺ | FAP⁺ | FAP⁺ |
| **Spatial distribution** | Present throughout stroma | In close proximity to vessels and vCAFs | In close proximity to vessels | Close to tumour-stroma border, often direct contact with tumour cells | Close to tumour-stroma border | Not detected | Around TLS | Not detected |

Summarising marker genes, anticipated biological function, classification as either activated or structural CAF, and the spatial distribution for each CAF type as described in this study.

Both our CAF classification system and naming scheme are intended as proposals, rather than strict definitions or descriptions of CAF types. We identified the CAF phenotypes based on unsupervised clustering of single-cell transcriptomics data, which requires choices to be made about the clustering resolution that defines meaningful cell types and/or states. Our choice of resolution was aimed at balancing over-clustering (i.e., defining too many clusters) and identifying rare but distinct cell types (e.g., dCAFs and rCAFs). In addition, we chose to merge two sets of clusters with highly overlapping differentially expressed genes (Supplementary Fig. 1b). While this conservative clustering choice should help guard against over-clustering due to technical effects and yielded biologically interpretable cell types (mCAFs and apCAFs), it remains possible that the merged clusters correspond to further subgroups of these CAF types.

In addition, naming cell types in a meaningful way that is also practically useful for the community comes with many challenges. We sought, when possible, to use already established names, when they are well grounded and when the described phenotypes matched our data. In other cases, we have proposed names that are our best interpretation of the functions associated with the top differentially expressed marker genes in each cell type. As such, our proposed naming scheme is based on previous studies in many laboratories, and we hope that it strikes a useful balance between pragmatism and biological meaning. Nevertheless, this naming scheme should be subject to revision based on functional experiments.

It is in any event likely that the CAF phenotypes we have identified represent a fluid spectrum of phenotypes with different functional features rather than fixed cellular states. Consistent with this, several of the phenotypes cluster closely together in the UMAP space, and clustering with subsets of markers rather than the entire measured transcriptome show small fluctuations in the distribution of CAF phenotypes. Time course experiments in model systems would be needed to study such relationships further. In addition, although we have defined each CAF phenotype based on differentially expressed genes that suggest a dominant biological feature or function, these cell types may harbour subgroups with different expression patterns and may well have more than one biological function. For example, although we defined mCAFs based on the expression of genes involved in matrix remodelling and production, they can also produce inflammatory factors such as cytokines and chemokines and some express genes known to facilitate adhesion and migration without it being their defining feature. Similarly, while tCAFs mainly express genes similar to those expressed in tumour cells, they also express MMPs and matrix proteins and may therefore also function to remodel the ECM.

Our identified phenotypes share markers with previously defined CAF phenotypes[12,15,16,21], especially those previously found in mouse models of breast cancer[15,16]. Cells we identified as vCAFs based on the expression of *MCAM*, *NOTCH3*, and *COL18A1* are analogous to the vascular CAFs defined in mouse, and the cells we define as dCAFs are analogous to cycling CAFs identified in mouse[15]. Inflammatory CAFs and immune regulatory CAFs were previously defined based on expression of *Il6*, *Cxcl12* and *Cxcl1* and *IL6* and *CXCL12*, respectively, in scRNA-seq data from mouse cancer models[7,16] and human cancer samples[7,14,16,21,42]. Although these patterns were recapitulated in our data, we defined a single category of *PLA2G2A⁺/CD34⁺/CFD⁺* iCAFs, since all the cells contained within this phenotype expressed genes actively involved in shaping the immune environment. Friedman et al. described ECM-CAFs based on the expression of *Fbn1*, which encodes fibronectin[16]. *Fbn1* is highly expressed by our mCAFs, but we found that *MMP11*, also used by Wu et al. to describe myofibroblast-like CAFs[21], was more strongly expressed in mCAFs. Finally, our apCAFs, defined by their expression of mRNAs encoding MHC-II molecules, were previously described by Elyada et al. in human and mouse PDAC[12].

Pericytes in human breast cancer, defined by the expression of *RGS5*, have previously been shown to have gene expression patterns

similar to those of CAFs[21]. Wu et al. also defined immature and developed perivascular-like cells[21], offering a potential trajectory from pericytes to vCAFs. Our vCAFs share features with both pericyte types but are more closely related to the developed perivascular-like cells in ref. 21. It remains to be determined whether pericytes are an independent cell type or whether they are a subset of specialised fibroblasts found in both healthy and diseased tissues.

rCAFs are closely related to vCAFs based on the UMAP. They express *CCL21* and *CCL19* and as such, are similar to reticular fibroblast cells found in lymphoid structures. *CCL19*-expressing CAFs have been previously described in cancer, usually as part of larger CAF groups[14,24], but a recent study labelled a similar CAF type as TLS CAFs[41]. We chose to define rCAFs as an individual category, despite their relative rarity, since we detected them mostly in close proximity to TLS in the IMC analysis.

We defined tCAFs based on the fact that their top differentially expressed genes are typically found expressed in tumour cells, for instance, *MME* (encoding for CD10), *NT5E* (encoding for CD73), *NDRG1* and ENO1, and gene set analysis revealed enrichment of the glycolysis hallmark pathway in this CAF type. Prior evidence also suggests that cells expressing these markers may have tumour-promoting and immunosuppressive effects; CD10 expression in tumour stroma and in CAFs has previously been associated with tumour stemness and chemoresistance in non-small cell lung cancer as well as with high tumour grades and poor patient survival[18,43–45]. The close spatial relationship of tCAFs with tumour cells suggests that these two cell types might interact with each other, which would be consistent with a tumour-promoting role. However, since our study is descriptive, further functional work will be needed to assess this hypothesis.

The fact that we were able to define almost all CAF phenotypes in datasets from multiple cancer types, both when analysed individually and when analysed as an integrated dataset, despite technical noise and likely biological differences, strongly suggests that CAF phenotypes are conserved across cancer types. A recent pan-cancer study has compared plasticity of fibroblasts and showed that distinct features are shared between the different tumour sites[23]. This may also extend to subtypes; for example, our breast cancer dataset mainly consisted of luminal B cancer, but the CAF phenotypes we defined were also present in samples of the other subtypes. Still, our data also indicate that different cancer types may be dominated by different CAF types. In the case of highly fibrotic PDAC, for instance, we were able to identify all CAF types but the main clusters were mCAFs and iCAFs.

We provide here a general CAF classification system that we have validated across cancer types; this panel identifies the phenotypic categories of vascular CAFs, matrix CAFs, immune CAFs, tumour-like CAFs, interferon-response CAFs, antigen-presenting CAFs, dividing CAFs, and reticular-like CAFs. We used our data to identify markers for each of these CAF types, for use in multiplex IMC. However, should the nature of follow-up studies require different markers (e.g., investigation of a particular functional subpopulation of CAFs), our dataset can also serve as a reference for choosing the most appropriate marker genes. In either context, our concepts and dataset should establish a basis for future studies of this important cell type.

## Methods
### Clinical samples
All clinical samples, as well as the corresponding clinical information, were collected after approval by the Cantonal Ethics Committee Zurich #2016-00215 as well as approval by the ethics committee at Rhenish Friedrich-Wilhelms University of Bonn #255/06 with written and informed consent from the patients who were not compensated.

All primary breast cancer samples included in this study were previously collected in collaboration with the Patients' Tumor Bank of Hope (PATH, Germany) and analysed by CyTOF[46], scRNA-seq, and IMC[25] for in-depth characterisation of tumour and immune landscapes.

All tissue and health-related data were collected under approval of the Ethics Committee Zurich (#2016-00215) and the faculty of medicine ethics committee at Friedrich-Wilhelms-University Bonn (#255/06). The fibroblast scRNA-seq data analysed in this study are part of a dataset previously generated by our laboratory[25]. The presented IMC data were specifically acquired for this study. For two of the 14 breast cancer samples for which scRNA-seq data were available, IMC measurements were not possible due to missing patient consent for FFPE-based analysis or due to low-quality FFPE material. Tumour subtypes in this study were defined as follows: Luminal A (ER$^+$ and/or PR$^+$, HER-2$^-$, Ki-67$^+$ <20%), Luminal B (ER$^+$ and/or PR$^+$, HER2$^-$, Ki-67 ≥20%), Luminal B-HER2$^+$ (ER$^+$ and/or PR$^+$, HER2$^+$), HER2$^+$ (ER$^-$, PR$^-$, and HER2$^+$), and triple negative (ER$^-$, PR$^-$ and HER2$^-$).

### Breast cancer−scRNA-seq dataset and fibroblast identification
The scRNA-seq dataset presented here was previously generated using the 10x Genomics platform, and the raw data pre-processing, quality control steps, and main cell type annotation have been described[25]. Briefly, gene-by-cell matrices were generated from the raw sequencing data using CellRanger (10x Genomics, v3.0.1) and subsequently transformed into Seurat objects (Seurat v3.0.2). After removing high-confidence doublets using the DoubletFinder Package, all Seurat objects were merged, and single cells with >7500 or <200 genes, with >75000 read counts, or with >20% of reads mapping to mitochondrial RNA were excluded. Highly variable genes were identified by the *sctransform* wrapper in Seurat and used to construct principal components. The principal components covering the highest variance in the dataset were used as input for graph-based clustering. Differential gene expression analysis was performed for the resulting clusters, and main cell types were annotated based on the cluster expression of established marker genes (*EPCAM* and *CDH1* for epithelial cells, *PECAM1* and *VWF* for endothelial cells, *PDGFRB* and *FAP* for fibroblasts, *CD3*, *CD4*, *CD8*, and *NCR1* for the T and NK cell fraction, *CD14*, *ITGAX* and *HLA-DRA* for myeloid cells, *MS4A2* for mast cells and basophils, *MS4A1* for B cells, and immunoglobulin-encoding genes for plasma cells). For the present study, all clusters annotated as "fibroblasts" in the original dataset were used for downstream analysis.

### Breast cancer−analysis of fibroblast clusters
All scRNA-seq analyses were done using R versions 4.1.2 and 4.1.3. We used the Seurat package (4.1.1) function *sctransform* to normalise and scale the data, using the varia"les "perce"t.mt" (mitochondrial genes), "percen".krt","and "percen".MGP" for regression (*KRT* and *MGP* were detected across all cell types in some samples due to contamination originating from apoptotic tumour cells). We used the *Seurat* harmony batch correction wrappers to reduce a visible sample effect. After running principal component analysis, the first 25 components were used for both graph-based clustering using the Seurat functions *FindNeighbours* and *FindClusters* as well as dimension reduction analysis such as UMAP. We used different resolutions in resolution steps of 0.1 for clustering and investigated the clustering hierarchy by plotting a clustree[47]. Use of results of clustering at resolution 0.4 resulted in a total of 12 clusters (numbered 0 through 11, Supplementary Fig. 2c and Supplementary Data 2). Differential gene expression per cluster was analysed using the *FindAllMarkers* function using *MAST* (version 1.20.0) testing (min.pct = 0.25, logfc.threshold = 0.25, two-sided); All *P* values were adjusted using Bonferroni correction (p_val_adj). Due to similar gene expression patterns, clusters 0 and 7 were unified as mCAFs, and clusters 8 and 10 were unified as apCAFs. Cluster 1 was assigned as iCAFs, cluster 2 as vCAFs, cluster 3 as pericytes, cluster 4 as tCAFs, cluster 5 as heat-shock protein-high tCAFs, cluster 6 as ifnCAFs, cluster 9 as rCAFs, and cluster 11 as dCAFs. Differential gene expression analysis was repeated for all final CAF clusters (Supplementary Data 3). The visual division between CAF types on the UMAP (Fig. 2b) was analysed and as differential gene expression analysis (Supplementary

Fig. 4) and feature plots (Fig. 2b) revealed that CAFs could be broadly differentiated by their *FAP* expression level, we used the clustering resolution of 0.1 and assigned clusters 0, 2, 3 and 4 as *FAP*+ and cluster 1 as *FAP* (later splitting up into vCAFs, pericytes and rCAFs, Supplementary Fig. 1c). Differential gene expression analysis was carried out to compare these two groups (Supplementary Data 4). The data was subclustered using only the final IMC panel genes to assess CAF type recovery with a 41-plex antibody panel (Supplementary Fig. 8b).

## Gene set enrichment analysis

We used the singleseq package (version 0.1.2.9000)[29] to run gene set enrichment analysis and compare the enrichment of hallmark pathways[28] between our defined CAF types (Supplementary Data 7).

## Analysis of validation datasets

The HNSCC, colon cancer, and PDAC datasets were downloaded from the NCBI Gene Expression omnibus (GSE103322, GSE132465, GSE154778 and GSE212966, respectively). The fibroblast subsets were then scaled and clustered as described above. For the HNSCC dataset, the original fibroblast identification was used. They were then clustered and the resolution of 0.6 was used for fibroblasts resulting in 10 clusters (0, vCAF; 1, vCAF; 2, iCAF; 3, tCAF; 4, Pericyte; 5, rCAF + apCAFs; 6, tCAF; 7, Pericyte; 8, other). The colon cancer dataset was clustered and cluster 6 at the resolution of 0.2 was identified as fibroblasts. Fibroblasts were then clustered at resolution 0.3 obtaining nine clusters (0, iCAF; 1, mCAF; 2, iCAF; 3, mCAF; 4, Pericyte; 5, mCAF; 6, other; 7, tCAF; 8, rCAF). Cells identified as "other" were excluded from subsequent analysis. For the lung cancer dataset, raw sequencing files (E-MTAB-6149, E-MTAB-6653) were downloaded and analysed using the *Cell-Ranger* pipeline (v6.0.0, refdata-gex-GRCh38-2020-A) to obtain gene-by-cell matrices before being analysed in a Seurat object. Fibroblasts were identified from both datasets (E-MTAB-6149, cluster 9 at resolution 0.2; E-MTAB-6653, cluster 12 at resolution 0.2) and merged into one Seurat object. The final clustering of the lung cancer fibroblasts was obtained at the resolution of 0.8 to yield 15 clusters (0, mCAF; 1, tCAF; 2, mCAF; 3, rCAF; 4, iCAF; 5, other; 6, apCAF; 7, iCAF; 8, pericyte; 9, iCAF; 10, epithelial; 11, other; 12, vCAF; 13, iCAF; 14, mitochondrial count high). The small PDAC dataset (GSE154778) all cells were clustered, and fibroblasts were identified in clusters 1 and 6 at resolution 0.1. Fibroblasts were then clustered, and resolution of 0.4 was chosen for the final cluster assignment resulting in a total of five clusters (0, iCAF; 1, mCAF; 2, apCAF; 3, pericytes; 4, other). For the bigger PDAC dataset (GSE212996) all cells were clustered, and fibroblasts were identified as clusters 2 and 4 at the resolution of 0.1. For fibroblasts, a clustering resolution of 0.6 was chosen, resulting in a total of 10 clusters (0, mCAF; 1, mCAF; 2, iCAF; 3, tCAF; 4, vCAF; 5, apCAF; 6, other; 7, pericyte; 8, iCAF; 9, other). All cells identified as other, epithelial, or mitochondrial high were excluded from subsequent analyses. The different clustering resolutions resulted from the varying sizes of the datasets and our aim to also detect the smaller subclusters.

## Dataset integration

All scRNA-seq datasets were integrated using the in-built *SelectIntegrationFeatures and FindIntegegrationAnchors* functions from the Seurat package. Batch correction was run using the harmony wrapper function for *Seurat*. Principle component analysis was performed, and the first 15 components were used in UMAP analysis. The integrated fibroblasts were analysed according to the steps previously described in the analysis of the breast cancer dataset (see above). Clustering at the resolution of 0.7 (Supplementary Data 5) resulted in 14 clusters (0, iCAF; 1, mCAF; 2, mCAF; 3, pericyte; 4, dCAF; 5, vCAF + rCAF; 6, mCAF; 7, tCAF; 8, iCAF; 9, other; 10, other; 11, dCAF; 12, apCAF; 13, other), differential gene expression for CAF types was repeated (Supplementary Data 6). When clustering the dataset using only the selected marker genes, this resulted in a total of 16 clusters at resolution 0.8

(0, iCAF; 1, mCAF; 2, mCAF; 3, rCAF + pericyte; 4, pericyte; 5, tCAF; 6, ifnCAF; 7, iCAF; 8, pericyte; 9, tCAF; 10, iCAF; 11, rCAF + pericyte; 12, iCAF; 13, iCAF; 14, dCAF; 15, apCAF) (Supplementary Fig. 6d, e).

## Tissue preparation and staining

Using a series of HistoClear and graded ethanol solutions, FFPE tissue sections were deparaffinised and rehydrated before antigen-retrieval in a decloaking chamber for 30 min at 95 °C using HIER buffer (pH 9.2). Tissues were blocked with 3% BSA for 45 min before being stained with the first metal-tagged antibody (anti-SMA, Supplementary Table 1) at room temperature for 4 h. After washing, the tissue was incubated with a fluorescently labelled anti-mouse (Abcam, AF 555, goat anti-mouse (H + L), catalogue number A21422; RRID_AB_2535844, polyclonal) for 1 h at room temperature before another washing step and final incubation with Hoechst (dilution 1:500) for 5 min. Afterwards, tissue was incubated with the remaining metal-tagged antibodies (Supplementary Table 1) at 4 °C overnight. After washing the next day, the samples were stained with an iridium DNA intercalator before being dried using pressured air.

## Immunofluorescence imaging

Whole slide scans were performed using a *Zeiss* AxioScan.Z1 with ×10 magnification. The resulting images were used to select between 6 and 8 stroma-rich areas, and, if present, TLS regions, of 1 mm$^2$ per patient for IMC imaging analysis (Supplementary Fig. 8d).

## Imaging mass cytometry

Brightfield scans of the slides were performed to detect the areas of interest to be measured by IMC. For subsequent imaging, the *Hyperion Imaging System*, coupled to a Helios time-of-flight mass cytometer was used at a laser intensity of 400 Hz (resolution at 1 μm). A compensation slide was also analysed to account for potential spill-over between the metals. The machine was tuned daily to account for machine performance variability. All markers were checked for their quality of staining (Supplementary Fig. 9).

## Analysis

Using the lab's analysis pipeline (github.com/BodenmillerGroup/ImcSegmentationPipeline [github.com/BodenmillerGroup/ImcSegmentationPipeline]), tiff files were generated from the raw data. Images were produced using *histoCAT*[48]. The tiff files were used for cellular segmentation based on nuclear and membrane markers using the *ilastik* software (version 1.3.3)[49]. First, fibroblasts were reduced to their nuclear core to reduce cellular overlap and signal falsification (Supplementary Fig. 10a, b). *CellProfiler* (v3.1.9)[50] was then used to generate cell masks and to calculate mean intensities of each marker per cell. Neighbours within a radius of 30 μm of each cell were also identified by *CellProfiler*. The single-cell data were analysed using R (v4.0.4); raw counts were censored, excluding the 99.99 percentile before being arc-sinh transformed using cofactor 1.

Graph-based clustering was carried out using the *Rphenoannoy* clustering algorithm (version 0.1.0). In the first clustering step, the dataset was divided into three compartments, tumour, immune, and stromal cells (Supplementary Fig. 8a, c). Cells identified as immune or stromal were then clustered individually, resulting in immune and stromal cell subtypes). CAFs were clustered using *FlowSOM* (version 2.2.0) clustering as integrated in the *CATALYST* (version 1.18.1) package[51]. We excluded HLA-DR from the CAF clustering as it resulted in too many false positive CAFs. This is due to the nature of HLA-DR expression on myeloid cells which are often widespread in the stroma and can thus cause false positive signal on other cell types. This was the only marker for which we observed this false positive signal.

CAFs were clustered using *FlowSOM* with a max k of 50 using SOM 50. vCAFs were identified as CD146+, CD31−/vWF−, FAP−, SMA+ cells, iCAFs were identified as CD34+, FAP+, CD31−/vWF− cells CD146$^{medium}$

cells. The overlap of CD146 for iCAFs and vCAFs can be explained by spill-over from CD34⁻ vCAFs. TCAFs were identified by their expression of CD73+, CD10+, PDPN+, FAP+, hypoxic tCAFs were identified by their positive expression of FAP, CD10 and CAIX. Heat-shock protein tCAFs were not identifiable with our panel. rCAFs showed higher than average expression of CCL21. ifnCAFs were identified by their expression of IDO. SMA CAFs showed high expression of SMA together with medium expression of PDGFR-b and collagens and fibronectin but did not stain positive for FAP. Lacking MMP11 as a marker for mCAFs, these were identified through negative selection, showing positive staining of FAP, SMA, Cadherin-11 (CDH-11), fibronectin and collagens as well as a slightly higher than average expression of PDPN but no expression of CD10, CD73, CD34 or CD146.The neighbourhood analysis was carried out analysing the 15 nearest cells in a defined radius of 25 μm of each cells' centroid using *imcRtools* (version 1.3.7). Differential abundance analysis was carried out using edgeR version 3.36.0 and diffcyt version 1.14.0.

### Image generation

Images shown in this study were generated using *histoCAT*-web and *cytomapper* (version 1.6.0).

### Reporting summary

Further information on research design is available in the Nature Portfolio Reporting Summary linked to this article.

## Data availability

We provide the raw data for all figures as well as the scripts used to generate all figures. IMC RAW data as well as single-cell objects of the IMC data and clustered scRNA-seq data objects have been deposited on zenodo under the following https://doi.org/10.5281/zenodo.5769017 [https://zenodo.org/record/7540604])[52]. The breast cancer RNA sequencing data are available with accession number E-MTAB-10607 on the ArrayExpress database at EMBL-EBI (www.ebi.ac.uk/arrayexpress). Publicly available datasets used in this study are available at the Gene Expression Omnibus with the following accession numbers: GSE132465 (colon cancer), GSE154778 PDAC (small), GSE212966 PDAC (big), GSE103322 (head-and-neck squamous cell carcinoma) and in ArrayExpress at EMBL-EBI under the accession numbers E-MTAB-6149 and E-MTAB-6653 (lung cancer).

## Materials availability

There were no new reagents generated in this study.

## Code availability

All analysis code is publicly available on Github (https://github.com/BodenmillerGroup/CAFclassification) and ref. 53 https://doi.org/10.5281/zenodo.7540622).

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

## Acknowledgements

B.B. was funded by an SNSF project grant, an NIH grant (UC4 DK108132), the CRUK IMAXT Grand Challenge, and the European Research Council (ERC) under the European Union's Horizon 2020 Program under the ERC grant agreement no. 866074 ("Precision Motifs"). B.B. was further founded by a SNSF SINERGIA project grant (#177208: Defining the identity and differentiation pathways of the immune-stimulating fibroblastic tumour stroma).

## Author contributions

L.C. generated the IMC dataset, analysed all scRNA-seq and IMC data, and wrote and revised the manuscript together with N.dS. S.T. generated the scRNA-seq dataset and provided input on the manuscript and analysis. T.A., C.L. and M.R. provided the samples as well as clinical input. N.dS. provided critical input on the manuscript. B.B. supervised the work and provided input at all levels.

## Competing interests

B.B. is a co-founder of Navignostics and a member of its board. The remaining authors declare no competing interests.
