## [Peer Review File · Nature Communications]

Reviewers' comments:

Reviewer #1 (Remarks to the Author):

This manuscript presents a cancer associated fibroblast classification based on sequencing of 14 human breast cancers, then validated on public datasets from other tumor types, including pancreatic cancer. The authors identify several known populations of fibroblasts, as well as describe some new populations based on prevalent gene expression. The authors then use highly multiplexed imaging mass spectrometry to visualize expression of genes associated with each fibroblast population in cancer samples, to evaluate their spatial distribution.

This manuscript follows a large number of studies using different gene expression parameters to identify specific fibroblast populations in cancer; such as PMID: 33981032. The use of imaging mass cytometry adds information as to protein expression; however, the study remains largely descriptive.

Higher magnification and high resolution images of the mass cytometry data would be useful to better evaluate the data from the images.

Given the number of articles defining fibroblast populations in healthy and diseased tissues, statements as to the novelty of this study should be moderated. In particular, the paragraph starting on line 56 states that previous studies of CAFs are cancer-type specific, but studies comparing fibroblast types across diseases exist: PMID: 33981032

PMID: 36333338, among others

Reviewer #2 (Remarks to the Author):

We commend the authors on the revised manuscript. They have addressed all of our original concerns.

Reviewer #3 (Remarks to the Author):

The revised manuscript by Cords et al includes improvements on the prior strategy to define a pan-CAF classification. However, many of the critiques raised by this reviewer, as well as from others, still remain. A few examples include:

- 1) The ultimate calling of CAF subtypes is still based on arbitrary and subjective decisions. As an example, cluster 0 and 7 are combined because they are “characterized by high levels of expression of genes encoding matrix proteins...”. Yet, based on the heatmap there are other genes that clearly are differentially expressed between cluster 0 and 7, so why combine these cells into a heterogenous cluster? The same reasoning is true for the merging of clusters 8 and 10. Also, how can the clusters that were arbitrarily combined exhibit overlapping genes in an analysis of differentially expressed genes? Any such merging of clusters should be objective and based on the dendrogram of the clustering (which by the way is missing throughout the paper). Also, were the DEG analysis rerun after the merging of some clusters?
- 2) In the description of vCAFs, it is stated that they do not express RGS5, whereas the heatmap clearly demonstrates prominent expression by a large proportion of these cells.
- 3) The designation of tumor promoting CAFs based on upregulation of oncogenic pathways is not biologically sound. Whereas activation of the K-Ras pathway or mutation/loss of p53 may be tumor-promoting in malignant cells, less is known about their effects in CAFs or other microenvironmental cell types. Without experimental evidence, this major concern still remains.
- 4) Similarly, the designation of CAFs into “activated” or “non-activated” is not robust or based on any experimental evidence. Is the division also reflected in the dendrogram of the clustering? The reason for this division is also not clear, since it is not used in any meaningful way.
- 5) The statement that similar pathways were upregulated in the GSEA of the pan-cancer dataset compared to the original discovery dataset is not true. As one example, the two distinguishing features of the vCAF subset in breast cancer, i.e. KRas down and PI3K signaling, are conspicuously absent from the pan-cancer analysis.
- 6) The marker profiles of CAF subsets in the imaging mass cytometry still does not match that of the mRNA expression defining the same subsets in the scRNAseq analysis. As an example, the authors state that mCAFs are detected by IMC by being PDPN+ (among other markers), but in the heatmap, mCAFs are negative for PDPN.

Reviewers' comments:

Reviewer #1 (Remarks to the Author):

This manuscript presents a cancer associated fibroblast classification based on sequencing of 14 human breast cancers, then validated on public datasets from other tumor types, including pancreatic cancer. The authors identify several known populations of fibroblasts, as well as describe some new populations based on prevalent gene expression. The authors then use highly multiplexed imaging mass spectrometry to visualize expression of genes associated with each fibroblast population in cancer samples, to evaluate their spatial distribution.

This manuscript follows a large number of studies using different gene expression parameters to identify specific fibroblast populations in cancer; such as PMID: 33981032. The use of imaging mass cytometry adds information as to protein expression; however, the study remains largely descriptive. Higher magnification and high resolution images of the mass cytometry data would be useful to better evaluate the data from the images.

We are showing the images at the highest resolution possible for IMC. We now provide the images at higher magnification images of the areas of interest (Fig 4c, reproduced below) and hope that this facilitates their evaluation.

Revised Figure 4c: Zoomed in images acquired with IMC showing the expression of key markers used in our classification system on the image level. The indicated CAF type is highlighted by arrows. CAFs are identified as follows: vCAFs, CD146; hypoxic CAFs: CDH-11, CA9; tmCAFs, SMA, CD10; IDO CAFs: IDO, SMA; iCAFs: aSMA, CD34; rCAFs: CCL21. PanCK indicates tumour cells, CD20 indicates B cells, Iridium (blue) indicates nuclei in all images. The scalebar is indicating 100 μ m.

Given the number of articles defining fibroblast populations in healthy and diseased tissues, statements as to the novelty of this study should be moderated. In particular, the paragraph starting on line 56 states that previous studies of CAFs are cancer-type specific, but studies comparing fibroblast types across diseases exist: PMID: 33981032 PMID: 36333338, among others.

We agree with the reviewer and have toned down these claims and now cite the mentioned studies.

Reviewer #2 (Remarks to the Author):

We commend the authors on the revised manuscript. They have addressed all of our original concerns.

We thank the review for the positive feedback.

Reviewer #3 (Remarks to the Author):

The revised manuscript by Cords et al includes improvements on the prior strategy to define a pan-CAF classification. However, many of the critiques raised by this reviewer, as well as from others, still remain. A few examples include:

1) The ultimate calling of CAF subtypes is still based on arbitrary and subjective decisions.

We respectfully disagree. Our classification and naming scheme is grounded in current knowledge and draws heavily on previously published studies. We discuss this in detail in the Discussion section. We provide references for the previous naming of several CAF types again below, studies are also cited in the manuscript. In many ways we unify with our scheme the various fragmented classifications out there in order to enable future work on CAFs by suggesting marker genes for generally identifiable and functionally interpretable CAF types. We have now explicitly stated this in the manuscript (lines 74-76).

Previous studies have defined mCAFs (or ECM CAFs) (Bartoschek *et al.*, 2018; Friedman *et al.*, 2020) , iCAFs (Wu *et al.*, 2020), dCAFs (Bartoschek *et al.*, 2018 (called cyclic CAFs), vCAFs (Bartoschek *et al.*, 2018), apCAFs (Elyada *et al.*, 2019; Friedman *et al.*, 2020; Kieffer *et al.*, 2020). In all cases, we now explicitly cite previous studies that have defined similar CAF types in the relevant section.

New categories we introduce that have not been proposed before are “rCAFs”, “IDO CAFs” and “tpCAFs”. The cells we call rCAFs have in previous work been included in the iCAF population (Kieffer *et al.*, 2020). However, the expression pattern of these cells (i.e., CCL19+ and CCL21+) distinguishes them from iCAFs in our study. Since these markers mimic fibroblastic reticular cells in the lymph node, we labeled them reticular CAFs. Our IMC findings further support this, since they show that trCAFs are often in close proximity to TLS, in contrast to iCAFs which are significantly enriched in non-TLS containing images (Figure 4f). In

fact, a recent study labels a similar CAF type (Grout *et al.*, 2022) as TLS CAFagain consistent with the term we use.

The cells we call tpCAFs have in previous work been associated with chemo-resistance (Su *et al.*, 2018). Given the spatial interaction of these cells with the tumour-stroma interface and given that they show similarly upregulated pathways as tumour cells, we called this population tumour promoting (tp)CAFs. However, as previously stated, and in response to the referee (point 3), we have renamed them tumour-mimicking CAFs.

Finally, the cells we call IDO CAFs have in previous work been described (Curran *et al.*, 2014) as CAFs secreting IDO in response to IFNs. The top differentially expressed genes of this cluster further include CXCL11, CXCL10, which are cytokines released in response to IFN stimulation, and GSEA analysis showed strong enrichment in IFN- α and IFN- γ response pathways, with enrichment scores higher than in other cell types where these pathways were also enriched (i.e., pericytes, dCAFs and rCAFs; see newly added Supp Table 7 for z scores). We thus named these cells IDO CAFs since IDO is a well-known marker for the cellular IFN response.

As an example, cluster 0 and 7 are combined because they are “characterized by high levels of expression of genes encoding matrix proteins...”. Yet, based on the heatmap there are other genes that clearly are differentially expressed between cluster 0 and 7, so why combine these cells into a heterogenous cluster? The same reasoning is true for the merging of clusters 8 and 10.

We combined clusters 0 and 7, and clusters 8 and 10, because it is meaningful given our current biological understanding of CAFs and as they shared differentially expressed marker genes such as MMP11 and collagens (for 0 and 7, yielding mCAFs) and MHC-II machinery related genes (for 8 and 10, yielding apCAFs). Please note that clusters 8 and 10 are highly related in the first place, and that this is also true, albeit more distantly, for clusters 0 and 7 (Fig. S1c, and reproduced below). Importantly, these shared differentially expressed genes belong to similar functional categories, and we focused on achieving a biologically meaningful grouping.

If we avoid merging these clusters and instead simply accept the clusters at a resolution of 0.4 in the dendrogram, the difficulty we face is that it is then very difficult/ to define separate markers that can be used to identify these cells in subsequent analyses. Given the strong biological overlap in the genes differentially expressed by these cells, we concluded that the better option would be to merge the clusters. Nevertheless, we acknowledge that the broad and functional classes of CAFs we defined may harbour subgroups with differences in gene expression patterns. This is now stated explicitly in the revised manuscript (line 410-413).

Also, how can the clusters that were arbitrarily combined exhibit overlapping genes in an analysis of differentially expressed genes?

This is precisely the point. Clusters such as 8 and 10, or 0 and 7, are similar to each other and therefore MUST have overlapping genes in the DGE analysis (which is done at the level of each cluster compared to all other clusters). This is in part the rationale for merging them. It is not unexpected in a clustering protocol that over-clustering can split up a functionally meaningful group into smaller groups. We do not claim here that the clusters we have selected and annotated represent the only way to look at the data or to understand CAF types. Rather we have sought a pragmatic and functionally/biologically motivated way to organize our scRNA-Seq data on this important cell type.

Any such merging of clusters should be objective and based on the dendrogram of the clustering (which by the way is missing throughout the paper).

It is incorrect that the dendrograms are missing from the paper. The dendrogram in the form of clustrees for the breast cancer scRNA-Seq dataset is in Supplementary Fig. 1c, and is reproduced here. We further show such clustrees for each validation dataset in supplementary figures 2a, 3a, 4a, 5a, 6a and 7a. The dendrograms for IMC analyses are included for all shown heatmaps (Supp Fig. 8a, and Fig. 4b). As mentioned above, the dendrograms for the breast cancer dataset (Supp. Fig 1c) shows that both clusters 8 and 10, and clusters 0 and 7, are related. We would also like to state that the selection of the level of cluster granularity and the generation of metaclusters in the light of known biology is commonly done in the scRNA-seq and IMC studies.

C

Supplementary figure 1c: Clustree showing hierarchical clustering for breast cancer fibroblasts with the black box indicating the chosen clustering resolution of 0.4.

Also, were the DEG analysis rerun after the merging of some clusters?

Yes, the results are shown in supplementary table 3. It might have been unclear before and we have now clearly stated this in the methods (line 601-602). GE results for clustering at resolution of 0.4 and for the final CAF types including merged clusters 0 and 7, 8 and 10 are provided (Supp. Tables 2&3).

2) In the description of vCAFs, it is stated that they do not express RGS5, whereas the heatmap clearly demonstrates prominent expression by a large proportion of these cells.

We thank the reviewer for pointing this out. Indeed, some cells within the vCAF cluster do express RGS5, but it is not a consistent marker as it is for pericytes, and it is therefore also not among the top differentially expressed genes for vCAFs. This is in line with fibroblasts characterised by genes involved in angiogenesis and NOTCH pathway that also partially expressed RGS5 and that clustered with pericytes identified in NSCLC by (Lambrechts *et al.*, 2018). Also, note that additional genes differentiate the vCAF cluster from pericytes, such as MHY11 and RERGL. We have revised the statement regarding RGS5 expression in vCAFs as follows: “A fraction of these cells showed expression of pericyte marker *RGS5* (Figure 2a, b; Supplementary Tables 2 and 3), but *RGS5* was not among the top differentially expressed genes of this cluster (Supp. Table 2, 3).”

3) The designation of tumor promoting CAFs based on upregulation of oncogenic pathways is not biologically sound. Whereas activation of the K-Ras pathway or mutation/loss of p53 may be tumor-promoting in malignant cells, less is known about their effects in CAFs or other microenvironmental cell types. Without experimental evidence, this major concern still remains.

We accept that this naming is speculative . We have renamed this cluster tumor-mimicking CAFs (tmCAFs) and avoid now any naming that is associated with function (also see next section)

4) Similarly, the designation of CAFs into “activated” or “non-activated” is not robust or based on any experimental evidence. Is the division also reflected in the dendrogram of the clustering? The reason for this division is also not clear, since it is not used in any meaningful way.

The division between these groups is indeed clear in the dendrogram (Figure S1c, see above). Activated (i.e., FAP+) CAFs correspond to clusters 0, 2, 3, 4 in the breast cancer dataset, while non-activated (i.e., FAP-) CAFs correspond to clusters 1, at the lowest clustering resolution. This division is further plainly visible on the UMAP (Fig. R1).

Fig. R1: UMAP of breast cancer fibroblasts coloured according to groups defined by FAP expression (left). Excerpt of Fig. 2b showing the FAP expression per cell (right).

However, we have now removed the “activated” versus “non-activated” terminology and label these clearly separated groups simply as FAP+ and FAP- CAFs, also in line with the literature (Yang *et al.*, 2016; Li *et al.*, 2017; Öhlund *et al.*, 2017; Costa *et al.*, 2018).

5) The statement that similar pathways were upregulated in the GSEA of the pan-cancer dataset compared to the original discovery dataset is not true. As one example, the two distinguishing features of the vCAF subset in breast cancer, i.e. KRas down and PI3K signaling, are conspicuously absent from the pan-cancer analysis.

The referee is fully correct that the results of the GSEA analyses on the original breast discovery dataset and the pan-cancer validation dataset are not identical, that is why we use “similar” and not “identical” in the manuscript. We do not think this is in any way unexpected given that the validation dataset includes multiple cancer types, and given that pathway definitions are in themselves imperfect. However, there are strong overlaps in the enriched pathways for the two datasets.

Comparing the top 5 enriched pathways in the breast cancer dataset shows high overlap among the enriched pathways (mCAF: 3/5, iCAF: 3/5, tpCAF: 4/5 apCAF:4/5, dCAF: 3/5, Revised Supplementary Table 7). Importantly, these almost always included 2 or 3 pathways corresponding to genes that we used in our functional annotation of the CAF type. The one

exception was iCAFs, where only a single pathway (IL6-JAK-STAT signaling) fell into this category, with complement signaling and Kras signaling being enriched, but lower on the list.

We have now revised the relevant text (line 253-262) in the manuscript as follows:

“Analysis of this integrated dataset yielded similar top differentially expressed genes for each CAF type as identified in the breast cancer dataset (Figure 3b, c, e; Supplementary Tables 5 and 6), and GSEA analysis showed that the top 5 enriched pathways in the breast cancer dataset were also enriched in the integrated dataset in 60-80% of cases (Supplementary Table 7). Although imperfect, we saw particularly strong overlap in the pathways that informed our functional annotation of CAF types (Figure 3f, Supp. Table 7). Specifically, the top 5 enriched pathways in the integrated dataset included hypoxia in tmCAFs, the EMT pathway and KRAS signalling in mCAFs, IL6-JAK-STAT signalling in iCAFs, allograft rejection in apCAFs, and E2F targets and the G2M checkpoint in dCAFs, all of which overlapped with top enriched pathways in the breast cancer dataset (Figure 3f).”

We now show the comparison of the top 20 enriched pathways in the breast cancer and integrated datasets in Supplementary Table 7 in the revised manuscript. We have also altered the markup of Fig. 3f (reproduced below) to make the extent of overlap clearer.

Revised figure 3f: Heatmap showing the results of gene set enrichment analysis for all defined cell types in the integrated dataset. Boxes indicate the overlap of the top 5 enriched hallmark pathways between the integrated validation and breast cancer dataset.

6) The marker profiles of CAF subsets in the imaging mass cytometry still does not match that of the mRNA expression defining the same subsets in the scRNA-seq analysis. As an example, the authors state that mCAFs are detected by IMC by being PDPN+ (among other markers), but in the heatmap, mCAFs are negative for PDPN.

Indeed, PDPN expression in mCAFs is low; note that it is higher however than in iCAFs or vCAFs, and it is also higher than expression of the markers used for negative selection of mCAFs (CD146, CD10, CD34) (Figure 4b, reproduced below). Nevertheless, we should have written this more clearly and we are sorry for the confusion. We also noticed that fibronectin and collagen are in fact included in the same imaging channel, which was also not clear previously. The revised sentence is now “Due to the lack of an antibody suitable for detection of MMP11, mCAFs were identified as PDPN^{low}/collagen-fibronectin^{high}/FAP⁺ cells that did not express CD10, CD34, or CD146”. We have carefully checked and revised the section to avoid similar lack of clarity regarding other markers used for identification of CAF types in IMC.

Figure 4b: Heatmap of marker expression of CAF clusters defined by IMC in breast tumour samples. The histogram indicates the square root of all cell numbers per cluster in each CAF type.

References

- Bartoschek, M. *et al.* (2018) 'Spatially and functionally distinct subclasses of breast cancer-associated fibroblasts revealed by single cell RNA sequencing', *Nature Communications* [Preprint]. Available at: <https://doi.org/10.1038/s41467-018-07582-3>.
- Costa, A. *et al.* (2018) 'Fibroblast Heterogeneity and Immunosuppressive Environment in Human Breast Cancer', *Cancer Cell* [Preprint]. Available at: <https://doi.org/10.1016/j.ccell.2018.01.011>.
- Curran, T.A. *et al.* (2014) 'IDO expressing fibroblasts promote the expansion of antigen specific regulatory T cells', *Immunobiology*, 219(1). Available at: <https://doi.org/10.1016/j.imbio.2013.06.008>.
- Elyada, E. *et al.* (2019) 'Cross-species single-cell analysis of pancreatic ductal adenocarcinoma reveals antigen-presenting cancer-associated fibroblasts', *Cancer Discovery* [Preprint]. Available at: <https://doi.org/10.1158/2159-8290.CD-19-0094>.
- Friedman, G. *et al.* (2020) 'Cancer-associated fibroblast compositions change with breast cancer progression linking the ratio of S100A4+ and PDPN+ CAFs to clinical outcome', *Nature Cancer* [Preprint]. Available at: <https://doi.org/10.1038/s43018-020-0082-y>.
- Grout, J.A. *et al.* (2022) 'Spatial Positioning and Matrix Programs of Cancer-Associated Fibroblasts Promote T-cell Exclusion in Human Lung Tumors', *Cancer Discovery*, 12(11). Available at: <https://doi.org/10.1158/2159-8290.CD-21-1714>.
- Kieffer, Y. *et al.* (2020) 'Single-cell analysis reveals fibroblast clusters linked to immunotherapy resistance in cancer', *Cancer Discovery*, 10(9), pp. 1330–1351.
- Lambrechts, D. *et al.* (2018) 'Phenotype molding of stromal cells in the lung tumor microenvironment', *Nature medicine*, 24(8), pp. 1277–1289.
- Li, H. *et al.* (2017) 'Reference component analysis of single-cell transcriptomes elucidates cellular heterogeneity in human colorectal tumors', *Nature Genetics*, 49(5). Available at: <https://doi.org/10.1038/ng.3818>.
- Öhlund, D. *et al.* (2017) 'Distinct populations of inflammatory fibroblasts and myofibroblasts in pancreatic cancer', *Journal of Experimental Medicine*, 214(3), pp. 579–596.
- Su, S. *et al.* (2018) 'CD10+GPR77+ Cancer-Associated Fibroblasts Promote Cancer Formation and Chemoresistance by Sustaining Cancer Stemness', *Cell* [Preprint]. Available at: <https://doi.org/10.1016/j.cell.2018.01.009>.
- Wu, S.Z. *et al.* (2020) 'Stromal cell diversity associated with immune evasion in human triple-negative breast cancer', *The EMBO Journal* [Preprint]. Available at: <https://doi.org/10.15252/embj.2019104063>.
- Yang, X. *et al.* (2016) 'FAP Promotes immunosuppression by cancer-associated fibroblasts in the tumor microenvironment via STAT3-CCL2 Signaling', *Cancer Research*, 76(14). Available at: <https://doi.org/10.1158/0008-5472.CAN-15-2973>.

REVIEWER COMMENTS

Reviewer #1 (Remarks to the Author):

This revised manuscript proposes a classification of cancer associated fibroblasts across tumor types. There is strong interest in CAFs and in understanding their characteristics. The main limitation of the current study is that it is purely descriptive.

Reviewer #3 (Remarks to the Author):

Cords et al have further revised the text and argued their standpoint in light of the remaining comments. With respect to the merging of clusters on the grounds that they have similar differentially expressed genes, this reviewer still does not agree that this is a valid and objective approach. The reasoning to merge clusters based on biological understanding is made clear by the authors. But this is still not reason enough to disregard the relationships of clusters based on the objective measure of the dendrogram/clustree. Indeed, whereas clusters 8 and 10 are merged at a lower resolution of the dataset, clusters 0 and 7 were only merged at the very lowest resolution (and parts of them keep merging and splitting up at higher resolutions). Thus, the authors introduce a comparative analysis of different sets of clusters at varying resolutions. This is not biologically meaningful, as argued by the authors, since clusters at different levels of resolution may represent different fundamental characteristics of the cells, e.g. cell type (origin) at the lower resolutions and cell state (context) at the higher resolutions.

In addition, the nomenclature proposed by the authors is inconsistent. Some clusters bear names that refer to the expression of classes of genes (matrix CAFs, inflammatory CAFs), others that refer to the localization of cells (vessel-associated CAFs), some that refer to similarities to other cell types (tumor-mimicking CAFs, antigen presenting CAFs, reticular-like CAFs), and yet others that refer to the expression of single genes (IDO CAFs). The field is not furthered by introduction of such disparate nomenclatures, but instead needs a unifying framework for classification of CAFs. And again, the naming of tumor-mimicking CAFs needs corroborating experimental data since the implications of the statement are broad (what does it mean that a CAF is "tumor-mimicking"?).

Moreover, the fact that "similar" but not "identical" gene sets are enriched in the clusters of the integrated analysis illustrates the point that the authors cannot draw direct parallels between clusters from different cancers based on this analysis. GSEA is not an accurate analysis tool to determine identity

of cellular clusters, since the expression of different genes within the same gene set can give rise to artificial similarities (e.g. the fact that cells in cluster A express genes 1, 2, and 3 of gene set X does not mean that they are identical/similar with the cells in cluster B that express genes 4, 5, and 6 of gene set X). Also, as illustrated by the revised heatmap provided by the authors, individual gene sets may be enriched by several clusters, again calling into question the strategy to base comparisons between specific clusters of cells on broadly defined gene sets of up to as many as 200 genes that may be enriched in several different subsets.

Reviewer #4 (Remarks to the Author): Expert in scRNA-seq analysis; arbitrating reviewer

The study by Cords et al, provides a classification scheme for cancer-associated fibroblasts. With scRNA-seq and multiplexed imaging mass cytometry, they have identified generalizable CAF subtypes that can be identified across multiple cancers. This study will provide a good starting point for other researchers to classify CAFs in their own datasets.

While reviewer #3 brings up good points about merging clusters, this reviewer does not necessarily think that these points lead to the loss of biological meaning. It must also be noted that clustering algorithms can also split clusters apart due to the quality of cells (i.e. high MT reads, low transcript count, etc.) despite being the same cell type. Moreover, conservatively calling clusters is likely the more prudent approach (to prevent the issue of over clustering) so long as they are substantiated by biological evidence. The authors have provided sufficient explanation regarding the known biological context of the various fibroblasts and their associated studies. Providing a carefully worded statement in the manuscript covering the points of reviewer #3 on merging clusters at varying resolutions would suffice.

This reviewer agrees with the point of reviewer #3 where the nomenclature can seem inconsistent. In particular, the IDO CAFs stick out compared to the other names and would for example be better fitted to “ifnCAF” for “interferon response” or what the authors would deem appropriate.

This reviewer finds it astonishing that even similar gene sets are called across different datasets. It’s highly unlikely that the same gene sets would be enriched in the CAF types across different datasets due to batch effects, sparsity of scRNA-seq, disease differences, etc. Thus, retaining any general characteristics across the board is quite substantial.

REVIEWER COMMENTS

Reviewer #1 (Remarks to the Author):

This revised manuscript proposes a classification of cancer associated fibroblasts across tumor types. There is strong interest in CAFs and in understanding their characteristics. The main limitation of the current study is that it is purely descriptive.

Reviewer #3 (Remarks to the Author):

Cords et al have further revised the text and argued their standpoint in light of the remaining comments. With respect to the merging of clusters on the grounds that they have similar differentially expressed genes, this reviewer still does not agree that this is a valid and objective approach. The reasoning to merge clusters based on biological understanding is made clear by the authors. But this is still not reason enough to disregard the relationships of clusters based on the objective measure of the dendrogram/clustree. Indeed, whereas clusters 8 and 10 are merged at a lower resolution of the dataset, clusters 0 and 7 were only merged at the very lowest resolution (and parts of them keep merging and splitting up at higher resolutions). Thus, the authors introduce a comparative analysis of different sets of clusters at varying resolutions. This is not biologically meaningful, as argued by the authors, since clusters at different levels of resolution may represent different fundamental characteristics of the cells, e.g. cell type (origin) at the lower resolutions and cell state (context) at the higher resolutions.

We acknowledge that clusters 0 and 7 may be different subgroups within the merged matrix CAF (mCAF) cluster. Top differentially expressed genes exclusive to cluster 7 (*MGP* and *BGN*) could suggest an association of this small subgroup of CAFs with cartilage.

Nevertheless, we note that cluster 7 (435 cells) is much smaller than cluster 0 (4,090 cells) and strongly overlaps with the larger cluster 0 in terms of its top differentially expressed genes (Supplementary Fig. 1b, reproduced below). Also, cluster 7 does not show strongly differentially expressed genes that are exclusive to this cluster (i.e., rather than being shared with cluster 0). This may indicate that cluster 7 is the result of over clustering (i.e., too many clusters), which as reviewer 4 points out can happen for technical reasons.

We had decided to merge clusters 0 and 7 in order to choose relatively conservative clusters that are also biologically meaningful, as also recognized by reviewer 4. We have retained the merging of clusters 0 and 7, and 8 and 10, in the revised manuscript. The process and rationale for merging was already described in the previous version. Based on the comments of referees 3 and 4 we now include a further paragraph about clustering choices in the discussion (lines 363-373), and state explicitly that the merged clusters may represent subgroups within mCAFs and apCAFs (lines 111-114; 190-191; 370-373).

Supplementary Figure 1b: Heatmap showing the top 10 differentially expressed genes for each cluster at the resolution of 0.4. Black boxes highlight clusters 0 and 7 as well as their top 10 differentially expressed genes.

In addition, the nomenclature proposed by the authors is inconsistent. Some clusters bear names that refer to the expression of classes of genes (matrix CAFs, inflammatory CAFs), others that refer to the localization of cells (vessel-associated CAFs), some that refer to similarities to other cell types (tumor-mimicking CAFs, antigen presenting CAFs, reticular-like CAFs), and yet others that refer to the expression of single genes (IDO CAFs). The field is not furthered by introduction of such disparate nomenclatures, but instead needs a unifying framework for classification of CAFs. And again, the naming of tumor-mimicking CAFs needs corroborating experimental data since the implications of the statement are broad (what does it mean that a CAF is "tumor-mimicking"?).

We agree with the reviewer that the nomenclature is slightly inconsistent in terms of the naming categories. Proposing a useful naming scheme is however not an easy task, especially since we wished to also respect established names for some cell types (specifically apCAFs¹⁻⁴ and iCAFs^{2,5-8}, but also to some extent mCAFs^{2,9,10} and vCAFs^{3,10}).

We have now made further efforts to consolidate the naming categories. We have changed IDO CAFs to interferon response CAFs (ifnCAFs) as suggested by referee 4, so that we no longer have any CAF type named based on a single marker. We have changed vessel-associated CAFs to vascular CAFs (still vCAFs) since there is some prior work that has used this term for the same cell type; this has eliminated one instance of a location category. We have changed tumour-mimicking CAFs (which we acknowledge could unintentionally give the impression of an active mimicry process in these cells) into tumour-like cells (tCAFs). This is based on the overlap in gene expression patterns of tCAFs with that of tumour cells and is the most neutral term we could identify. We also include a speculative paragraph in the discussion about the markers expressed by tCAFs having been previously associated with the promotion of tumour growth (lines 432-438).

We thus now have a combination of established names (iCAFs, apCAFs, vCAFs, mCAFs), and names based either on a suggested functional association (mCAFs, ifnCAFs, dCAFs, but also iCAFs, apCAFs, vCAFs) or on similarity to other cell types (tCAFs, reticular CAFs). While this may not be a perfect strategy, the proposed scheme strikes a balance between pragmatism and consistency. We thank the referees for their ongoing attention to this point.

Importantly however, we do not intend to suggest that this is the only or the best naming scheme possible. Certainly, further studies and functional experiments may result in these names being changed or modified. We now include a new paragraph in the discussion (lines 375-383) explicitly discussing the challenges associated with naming cell types, outlining the decisions we have made in order to propose a useful scheme, and stating that further functional data will be needed to refine or confirm the names that we have proposed.

Moreover, the fact that "similar" but not "identical" gene sets are enriched in the clusters of the integrated analysis illustrates the point that the authors cannot draw direct parallels between clusters from different cancers based on this analysis. GSEA is not an accurate analysis tool to determine identity of cellular clusters, since the expression of different genes within the same gene set can give rise to artificial similarities (e.g. the fact that cells in cluster A express genes 1, 2, and 3 of gene set X does not mean that they are identical/similar with the cells in cluster B that express genes 4, 5, and 6 of gene set X). Also, as illustrated by the

revised heatmap provided by the authors, individual gene sets may be enriched by several clusters, again calling into question the strategy to base comparisons between specific clusters of cells on broadly defined gene sets of up to as many as 200 genes that may be enriched in several different subsets.

The reviewer is correct that GSEA cannot be used to determine the identity of cellular clusters, and we do not use GSEA in this way. We defined CAF types based on differential gene expression as measured with scRNA-seq, further assessed by protein level imaging with IMC. Naming of these clusters, as well as choices about cluster merging, were based on a literature-driven biological interpretation of the top differentially expressed genes. The GSEA was merely used to further investigate the profile of the identified CAF types and to assess whether these functional annotations based on prior knowledge were consistent with our proposed naming scheme. This analysis was proposed by a previous reviewer, and we see it as a useful orthogonal assessment of our defined clusters, but no more than that.

It is not at all surprising that the pathways enriched in different CAF types are not identical between cancer types. As referee 4 points out, given technical noise (batch effects and data sparsity) and likely disease differences it is indeed surprising that there is considerable overlap, strongly supporting that there are similarities between the CAF types in these data sets corresponding to different diseases. We note as well that genes like *FAP* or *PDPN* overlap between the different CAF types, which would further confound any specific signals.

Reviewer #4 (Remarks to the Author): Expert in scRNA-seq analysis; arbitrating reviewer
The study by Cords et al, provides a classification scheme for cancer-associated fibroblasts. With scRNA-seq and multiplexed imaging mass cytometry, they have identified generalizable CAF subtypes that can be identified across multiple cancers. This study will provide a good starting point for other researchers to classify CAFs in their own datasets.

We thank the reviewer for their supportive comments.

While reviewer #3 brings up good points about merging clusters, this reviewer does not necessarily think that these points lead to the loss of biological meaning. It must also be noted that clustering algorithms can also split clusters apart due to the quality of cells (i.e. high MT reads, low transcript count, etc.) despite being the same cell type. Moreover, conservatively calling clusters is likely the more prudent approach (to prevent the issue of over clustering) so long as they are substantiated by biological evidence. The authors have provided sufficient explanation regarding the known biological context of the various fibroblasts and their associated studies. Providing a carefully worded statement in the manuscript covering the points of reviewer #3 on merging clusters at varying resolutions would suffice.

We thank the reviewer for pointing out the value of conservative clustering and the technical reasons that clusters may be split. We had already clearly stated our rationale for merging clusters in the previous version of the manuscript and, based on these comments, we have added a further paragraph about clustering choices to the discussion (lines 363-

373), and state explicitly that merged clusters may represent subgroups within mCAFs and apCAFs (lines 370-373, see also response to reviewer 3).

This reviewer agrees with the point of reviewer #3 where the nomenclature can seem inconsistent. In particular, the IDO CAFs stick out compared to the other names and would for example be better fitted to “ifnCAF” for “interferon response” or what the authors would deem appropriate.

We agree with reviewers 3 and 4 that the nomenclature is slightly inconsistent in terms of the naming categories. Proposing a useful naming scheme is however not an easy task, especially since we wished to also respect established names for some cell types (specifically apCAFs and iCAFs, but also to some extent mCAFs).

We have now made further efforts to consolidate the naming categories, including changing IDO CAFs to interferon response CAFs (ifnCAFs) as suggested by the reviewer. Please see the response to reviewer 3 for a full description of our changes. We now have a combination of established names (iCAFs, apCAFs, vCAFs, mCAFs), and names based either on a suggested functional association (mCAFs, ifnCAFs, dCAFs, but also iCAFs, apCAFs, vCAFs) or on similarity to other cell types (tCAFs, reticular CAFs). While this may not be a perfect strategy, the proposed scheme strikes a balance between pragmatism and consistency. We thank the referees for their ongoing attention to this point.

Importantly however, we do not intend to suggest that this is the only or the best naming scheme possible. Certainly, further studies and functional experiments may result in these names being changed or modified. We now include a new paragraph in the discussion (lines 375-383) explicitly discussing the challenges associated with naming cell types, outlining the decisions we have made in order to propose a useful scheme, and fact that further functional data will be needed to refine or confirm the names that we have proposed.

This reviewer finds it astonishing that even similar gene sets are called across different datasets. It's highly unlikely that the same gene sets would be enriched in the CAF types across different datasets due to batch effects, sparsity of scRNA-seq, disease differences, etc. Thus, retaining any general characteristics across the board is quite substantial.

We agree. We do not, as suggested by reviewer 3, use the GSEA to identify cell clusters, but merely as an orthogonal analysis of our clusters and the proposed naming scheme. We were encouraged to find that there was some consistency between the annotated pathways and the names we had proposed based on the top differentially expressed genes in breast cancer, and indeed that there were even similarities between enriched pathway annotations in different cancer types.

As mentioned also in our response to reviewer 3, we do not think it at all surprising that the enriched pathways are not identical between data sets, for the reasons reviewer 4 highlights here, but also because there are shared genes expressed across all CAF types, further confounding the enrichment signal, and because any such enrichment analysis is based on prior knowledge that will itself be incomplete.

Overall, we share this reviewer's position. We think the fact that there is similarity in enriched gene sets across data sets/cancer types, but also, and importantly, that there is similarity in the defined clusters, in the top differentially expressed genes, and in our defined marker genes, strongly supports that these CAF types are - at least in a broad sense - general.

We thank the referee for their input and their support that this study will serve as a starting point for others to classify CAFs in their data.

References

1. Luo, H. *et al.* Pan-cancer single-cell analysis reveals the heterogeneity and plasticity of cancer-associated fibroblasts in the tumor microenvironment. *Nat Commun* **13**, 6619 (2022).
2. Yu, Z. *et al.* Single-cell sequencing reveals the heterogeneity and intratumoral crosstalk in human endometrial cancer. *Cell Prolif* **55**, (2022).
3. Martin-Serrano, M. A. *et al.* Novel microenvironment-based classification of intrahepatic cholangiocarcinoma with therapeutic implications. *Gut* **72**, (2023).
4. Kerdidani, D. *et al.* Lung tumor MHCII immunity depends on in situ antigen presentation by fibroblasts. *Journal of Experimental Medicine* **219**, (2022).
5. Chen, Z. *et al.* Single-cell RNA sequencing highlights the role of inflammatory cancer-associated fibroblasts in bladder urothelial carcinoma. *Nat Commun* **11**, 1–12 (2020).
6. Bernard, V. *et al.* Single-cell transcriptomics of pancreatic cancer precursors demonstrates epithelial and microenvironmental heterogeneity as an early event in neoplastic progression. *Clinical Cancer Research* **25**, 2194–2205 (2019).
7. Wu, S. Z. *et al.* Stromal cell diversity associated with immune evasion in human triple-negative breast cancer. *EMBO J* (2020) doi:10.15252/emj.2019104063.
8. Kieffer, Y. *et al.* Single-cell analysis reveals fibroblast clusters linked to immunotherapy resistance in cancer. *Cancer Discov* **10**, 1330–1351 (2020).
9. Zhang, M. *et al.* Single-cell transcriptomic architecture and intercellular crosstalk of human intrahepatic cholangiocarcinoma. *J Hepatol* **73**, 1118–1130 (2020).
10. Bartoschek, M. *et al.* Spatially and functionally distinct subclasses of breast cancer-associated fibroblasts revealed by single cell RNA sequencing. *Nat Commun* (2018) doi:10.1038/s41467-018-07582-3.

REVIEWERS' COMMENTS

Reviewer #3 (Remarks to the Author):

Differences of opinion are best settled openly and by the scientific community, and not through the peer review system. The authors are commended on their persistence.

Reviewer #4 (Remarks to the Author):

The manuscript has substantially benefited from the changes that Cords et al implemented in this revision round. The additional paragraph explaining the rationale behind their clustering approach puts things into a more conservative perspective, where future studies could potentially build on what this study has done, and is appreciated by this reviewer. No further comments.